# Revisiting Hierarchical Approach for Persistent Long-Term Video Prediction

**Wonkwang Lee**[1]**, Whie Jung**[1]**, Han Zhang**[2]**, Ting Chen**[2]**, Jing Yu Koh**[2]**,**
**Thomas Huang**[3]**, Hyungsuk Yoon**[4]**, Honglak Lee**[3,5]*****Seunghoon Hong**[1]
[1]KAIST, [2]Google Research, [3]University of Michigan, [4]MOLOCO, [5]LG AI Research
{wonkwang.lee, whieya, seunghoon.hong}@kaist.ac.kr
{zhanghan, iamtingchen, jykoh}@google.com
thomaseh@umich.edu
hyungsuk@molocoads.com
honglak@eecs.umich.edu

## Abstract

Learning to predict the long-term future of video frames is notoriously challenging due to inherent ambiguities in the distant future and dramatic amplifications of prediction error through time. Despite the recent advances in the literature, existing approaches are limited to moderately short-term prediction (less than a few seconds), while extrapolating it to a longer future quickly leads to destruction in structure and content. In this work, we revisit hierarchical models in video prediction. Our method predicts future frames by first estimating a sequence of semantic structures and subsequently translating the structures to pixels by video-to-video translation. Despite the simplicity, we show that modeling structures and their dynamics in the discrete semantic structure space with a stochastic recurrent estimator leads to surprisingly successful long-term prediction. We evaluate our method on three challenging datasets involving *car driving* and *human dancing*, and demonstrate that it can generate complicated scene structures and motions over a very long time horizon (*i.e.,* thousands frames), setting a new standard of video prediction with orders of magnitude longer prediction time than existing approaches. Full videos and codes are available at https://1konny.github.io/HVP/.

## 1 Introduction

Video prediction aims to generate future frames conditioned on a short video clip. It has received much attention in recent years as forecasting the future of visual sequence is critical in improving the planning for model-based reinforcement learning (Finn et al., 2016; Hafner et al., 2019; Ha & Schmidhuber, 2018), forecasting future event (Hoai & Torre, 2013), action (Lan et al., 2014), and activity (Lan et al., 2014; Ryoo, 2011). To make it truly beneficial for these applications, video prediction should be capable of forecasting *long-term* future. Many previous approaches have formulated video prediction as a conditional generation task by recursively synthesizing future frames conditioned on the previous frames (Vondrick et al., 2016; Tulyakov et al., 2018; Denton & Fergus, 2018; Babaeizadeh et al., 2018; Castrejon et al., 2019; Villegas et al., 2019). Despite their success in short-term forecasting, however, none of these approaches have been successful in synthesizing convincing long-term future, due to the challenges in modeling complex dynamics and extrapolating from short sequences to much longer future. As the prediction errors easily accumulate and amplify through time, the quality of the predicted frames quickly degrades over time.

One way to reduce the error propagation is to extrapolate in a low dimensional structure space instead of directly estimating pixel-level dynamics in a video. Therefore, many hierarchical modeling approaches are proposed (Villegas et al., 2017b; Wichers et al., 2018; Liang et al., 2017; Yan et al., 2018; Walker et al., 2017; Kim et al., 2019). These approaches first generate a sequence using a low-dimensional structure representation, and subsequently generate appearance conditioned on the predicted structures. Hierarchical approaches are potentially promising for long-term prediction since learning structure-aware dynamics allows the model to generate semantically accurate motion and content in the future. However, previous approaches often employed too specific and incomprehensive structures such as human body joints (Villegas et al., 2017b; Yan et al., 2018; Yang et al., 2018; Walker et al., 2017; Kim et al., 2019) or face landmarks (Yan et al., 2018; Yang et al., 2018).

---

*The author contributed to this work while at Google Research.

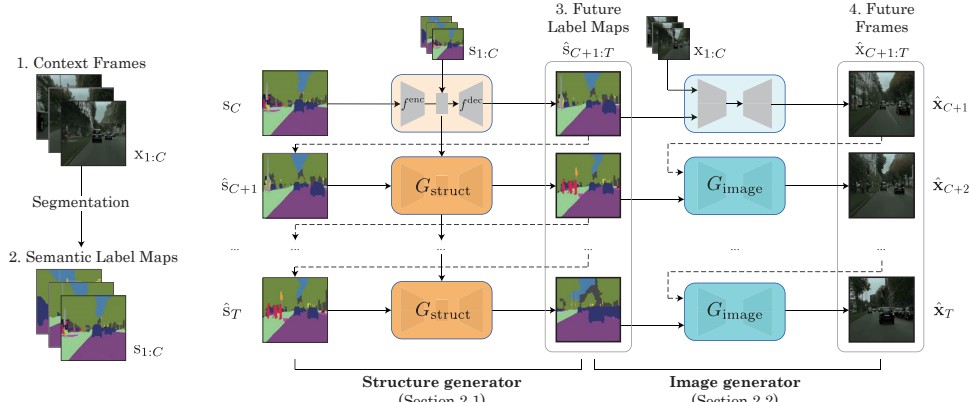

Figure 1: The overall framework of the proposed hierarchical approach. Given the context frames and label maps extracted by the segmentation network, our model predicts the future frames by estimating the semantic label maps using a stochastic sequence estimator (Section 2.1) and converting the predicted labels to RGB frames by using a conditional image sequence generator (Section 2.2).

Moreover, they made oversimplified assumptions of the future by using a deterministic loss or assuming homogeneous content. We therefore argue that the benefit of hierarchical models has been underestimated and their impact on long-term video prediction has not been properly demonstrated.

In this paper, we propose a hierarchical model with a general structure representation (i.e., *dense semantic label map*) for long-term video prediction on complex scenes. We abstract the scene as categorical semantic labels for each pixel, and predict the motion and content change in this label space using a variational sequence model. Given the context frames and the predicted label maps, we generate the textures by translating the sequence of label maps to the RGB frames. As dense label maps are generic and universal, we can learn comprehensive scene dynamics from object motion to even dramatic scene change. We can also capture the multi-modal dynamics in the label space with the stochastic prediction of the variational sequence model. Our experiments demonstrate that we can generate a surprisingly long-term future of videos, from driving scenes to human dancing, including the complex motion of multiple objects and even an evolution of the content in a distant future. We also show that predicted frame quality is preserved through time, which enables *persistent* future prediction virtually near-infinite time horizon. For scalable evaluation of long-term prediction at this scale, we also propose a novel metric called *shot-wise FVD*, which enables the evaluation of spatio-temporal prediction quality without ground-truths and is consistent with human perception.

## 2 METHOD

Given the context frames $\mathbf{x}_{1:C} = \{\mathbf{x}_1, \mathbf{x}_2, ..., \mathbf{x}_C\}$, our goal is to synthesize the future frames $\mathbf{x}_{C+1:T} = \{\mathbf{x}_{C+1}, \mathbf{x}_{C+2}, ..., \mathbf{x}_T\}$ up to an arbitrary long-term future $T$. Let $\mathbf{s}_t \in \mathbb{R}^{N \times H \times W}$ denote a dense label map of the frame $\mathbf{x}_t$ defined over $N$ categories, which is inferred by the pre-trained semantic segmentation model[1]. Then given the context frames $\mathbf{x}_{1:C}$ and the label maps $\mathbf{s}_{1:C}$, our hierarchical framework synthesizes the future frames $\hat{\mathbf{x}}_{C+1:T}$ by the following steps.

- A structure generator takes the context label maps as inputs and produces a sequence of the future label maps by $\hat{\mathbf{s}}_{C+1:T} \sim G_{\text{struct}}(\mathbf{s}_{1:C}, \mathbf{z}_{1:T})$, where $\mathbf{z}_t$ denotes the latent variable encoding the stochasticity of the structure.

- Given the context frames and the predicted structures, an image generator produces RGB frames by $\hat{x}_{C+1:T} \sim G_{\text{image}}(\mathbf{x}_{1:C}, \{\mathbf{s}_{1:C}, \hat{\mathbf{s}}_{C+1:T}\})$.

Figure 1 illustrates the overall pipeline. Note that there are various factors beyond motions that make spatio-temporal variations in the label maps, such as an emergence of new objects, partial observability, or even dramatic scene changes by the global camera motion (*e.g.*, panning). By learning to model these dynamics in the semantic labels with the stochastic sequence estimator and conditioning the video generation with the estimated labels, the proposed hierarchical model can synthesize convincing frames into the very long-term future. Below, we describe each component.

---

[1] We employ the pre-trained network (Zhu et al., 2019) on *still images* to obtain segmentations in videos.

## 2.1 SEQUENTIAL DENSE STRUCTURE GENERATOR

Our structure generator models dynamics in label maps by $p(\mathbf{s}_{\leq T}|\mathbf{z}_{\leq T}) = \prod_{t=1}^{T} p(\mathbf{s}_t|\mathbf{s}_{<t}, \mathbf{z}_{\leq t})$. We employ a sequence model proposed by Denton & Fergus (2018) since it (1) provides a probabilistic framework to handle stochasticity in structures and (2) can easily incorporate discrete sequences. Specifically, we optimize $\beta$-VAE objective (Higgins et al., 2017; Denton & Fergus, 2018) as below:

$$\sum_{t=1}^{T} \mathbb{E}_{q_\phi(\mathbf{z}_{\leq T}|\mathbf{s}_{\leq T})}[\log p_\theta(\mathbf{s}_t|\mathbf{z}_{\leq t}, \mathbf{s}_{<t})] - \beta D_{KL}(q_\phi(\mathbf{z}_t|\mathbf{s}_{\leq t})||p_\psi(\mathbf{z}_t|\mathbf{s}_{<t})). \tag{1}$$

where $q_\phi(\mathbf{z}_{\leq T}|\mathbf{s}_{\leq T})$ is the approximated posterior distribution and $p_\psi(\mathbf{z}_t|\mathbf{s}_{<t})$ is the prior distribution. At each step, it learns to represent and reconstruct each label map $\mathbf{s}_t$ through CNN-based encoder $f^{\mathrm{enc}}$ and decoder $f^{\mathrm{dec}}$, respectively, while the temporal dependency of the label maps is modeled stochastically by the two LSTMs as follows:

$$\begin{aligned}
\mu_\phi(t), \sigma_\phi(t) &= \mathrm{LSTM}_\phi(\mathbf{h}_t) & \text{where} \quad \mathbf{h}_t &= f^{\mathrm{enc}}(\mathbf{s}_t), \\
\mathbf{z}_t &\sim \mathcal{N}(\mu_\phi(t), \sigma_\phi(t)), \\
\mathbf{g}_t &= \mathrm{LSTM}_\theta(\mathbf{h}_{t-1}, \mathbf{z}_t) & \text{where} \quad \mathbf{h}_{t-1} &= f^{\mathrm{enc}}(\mathbf{s}_{t-1}), \\
\mathbf{s}_t &= f^{\mathrm{dec}}(\mathbf{g}_t),
\end{aligned} \tag{2}$$

where $\mathrm{LSTM}_\theta$ and $\mathrm{LSTM}_\phi$ respectively approximate the generative and the posterior distributions recurrently up to the time step $t$. Unlike Denton & Fergus (2018); Villegas et al. (2019) that exploit skip-connection from the last observed context frame during both training and testing time, we skip hidden representations of the encoder to the decoder at every time step during testing to handle long-term dynamics in structure as in (Villegas et al., 2017a; Finn et al., 2016).

During training, we apply teacher-forcing by feeding ground-truth label maps and sampling the latent $\mathbf{z}_t$ from the posterior distribution $\mathcal{N}(\mu_\phi(t), \sigma_\phi(t)) = \mathrm{LSTM}_\phi(f^{\mathrm{enc}}(\mathbf{s}_t))$. During inference, we recursively generate label maps by (1) sampling $\mathbf{z}_t$ from the prior distribution $\mathcal{N}(\mu_\psi(t), \sigma_\psi(t)) = \mathrm{LSTM}_\psi(f^{\mathrm{enc}}(\hat{\mathbf{s}}_{t-1}))$, (2) producing the frame $\tilde{\mathbf{s}}_t$ through the decoder, and (3) discretizing the predicted label map by taking pixel-wise maximum. Such discretization provides additional merits for being more robust against error propagation than continuous structures such as 2D keypoints (Villegas et al., 2017b; Yan et al., 2018; Yang et al., 2018) or optical flow (Walker et al., 2017).

**Extension to object boundary prediction** When the pre-trained instance-wise segmentation model is available, we can optionally extend the structure generator to jointly predict the object boundary maps. Such structure can add a notion of object instance, and is useful to improve the image generator in sequences with many occluding objects. Let $\mathbf{e}_t \in \{0, 1\}^{H \times W}$ denotes the object boundary map at frame $\mathbf{x}_t$. Then we train the conditional generator $\hat{\mathbf{e}}_t = G_{\mathrm{edge}}(\hat{\mathbf{s}}_t)$ that produces a boundary map given the label map for each frame[1]. We train $G_{\mathrm{edge}}$ using the conditional GAN objective to match the generated distribution to the joint distribution of the real label and real boundary maps $p(\mathbf{s}_t, \mathbf{e}_t)$. The boundary and label maps are then combined as the output of the structure generator $\tilde{\mathbf{s}}_t = [\hat{\mathbf{s}}_t, \hat{\mathbf{e}}_t]$, and are used as inputs to the image generator described below.

## 2.2 STRUCTURE-CONDITIONAL PIXEL SEQUENCE GENERATOR

Given a sequence of the structures and the context frames, the image generator learns to model the conditional distribution of the RGB frames by $p(\mathbf{x}_{\leq T}|\mathbf{s}_{\leq T}) = \prod_{t=1}^{T} p(\mathbf{x}_t|\mathbf{x}_{<t}, \mathbf{s}_{\leq t})$. We formulate this task as a video-to-video translation problem, and employ a state-of-the-art conditional video generation model (Wang et al., 2018). Specifically, the video synthesis network $F$ in Wang et al. (2018) consists of three main components; the generator $H$, occlusion mask predictor $M$, and optical flow estimator $W$, which are combined to generate each frame $\hat{\mathbf{x}}_t$ by the following operation:

$$\hat{\mathbf{x}}_t = F(\hat{\mathbf{x}}_{t-\tau:t-1}, \mathbf{s}_{t-\tau:t}) = (1 - \mathbf{m}_t) \odot \hat{\mathbf{w}}_{t-1} + \mathbf{m}_t \odot \mathbf{h}_t \tag{3}$$

where $\hat{\mathbf{w}}_{t-1} = W(\hat{\mathbf{x}}_{t-\tau:t-1}, \mathbf{s}_{t-\tau:t})$ is the warped previous frame $\hat{\mathbf{x}}_{t-1}$ using the estimated optical flow, $\mathbf{h}_t = H(\hat{\mathbf{x}}_{t-\tau:t-1}, \mathbf{s}_{t-\tau:t})$ is the hallucinated frame at $t$, and $\mathbf{m}_t = M(\hat{\mathbf{x}}_{t-\tau:t-1}, \mathbf{s}_{t-\tau:t})$ is the soft occlusion mask blending $\hat{\mathbf{w}}_{t-1}$ and $\mathbf{h}_t$. Unlike models synthesizing future frames by transforming the context frames $\mathbf{x}_C$ (Villegas et al., 2019; 2017b; Yan et al., 2018; Tulyakov et al., 2018), this

---

[1] We do not feed the predicted boundary map as input to the structure generator since it makes the structure generator prune to error propagation thus prevents extrapolation to long-term future.

model is appropriate to synthesize the long-term future since it can handle both transformation of the existing objects (via $\hat{\mathbf{w}}_{t-1}$) and synthesis of the emerging objects (via $\mathbf{h}_t$).

To ensure both frame-level and video-level generation quality, the video synthesis network $F$ is trained against a conditional image discriminator $D_I$ and a conditional video discriminator $D_V$ through adversarial learning by

$$\mathcal{L}_I(F, D_I) = \mathbb{E}_{\phi_I(\mathbf{x}_{\leq T}, \mathbf{s}_{\leq T})}[\log D_I(\mathbf{x}_i, \mathbf{s}_i)] + \mathbb{E}_{\phi_I(\hat{\mathbf{x}}_{\leq T}, \mathbf{s}_{\leq T})}[\log (1 - D_I(\hat{\mathbf{x}}_i, \mathbf{s}_i))], \qquad (4)$$

$$\mathcal{L}_V(F, D_V) = \mathbb{E}_{\phi_V(\mathbf{w}_{<T}, \mathbf{x}_{\leq T}, \mathbf{s}_{\leq T})}[\log D_V(\mathbf{x}_{i-1:i-\tau'}, \mathbf{w}_{i-2:i-\tau'})] +$$
$$\mathbb{E}_{\phi_V(\mathbf{w}_{<T}, \hat{\mathbf{x}}_{\leq T}, \mathbf{s}_{\leq T})}[\log (1 - D_V(\hat{\mathbf{x}}_{i-1:i-\tau'}, \mathbf{w}_{i-2:i-\tau'}))], \qquad (5)$$

where $\mathcal{L}_I(F, D_I)$ and $\mathcal{L}_V(F, D_V)$ are frame-level and video-level adversarial losses, respectively. For efficient training, we follow Wang et al. (2018) to adopt sampling operators $\phi_I(\mathbf{x}_{\leq T}, \mathbf{s}_{\leq T}) = (\mathbf{x}_i, \mathbf{s}_i)$ and $\phi_V(\mathbf{w}_{<T}, \mathbf{x}_{\leq T}, \mathbf{s}_{\leq T}) = (\mathbf{w}_{i-2:i-\tau'}, \mathbf{x}_{i-1:i-\tau'}, \mathbf{s}_{i-1:i-\tau'})$ for frame and video-level adversarial learning objectives, respectively, where $i$ is an integer sampled from $\mathrm{U}(1, T)$ and $\mathrm{U}(\tau' + 1, T + 1)$ for frame sampling operator $\phi_I$ and video sampling operator $\phi_V$, respectively. Then the final learning objective is formulated by

$$\min_F \max_{D_I, D_V} \mathcal{L}_I(F, D_I) + \mathcal{L}_V(F, D_V). \qquad (6)$$

During training, the model is trained to estimate RGB frames from the GT structures, while the predictions from the structure generator are used during testing. See Appendix C.3 for more details.

## 3 RELATED WORK

The task of predicting future video frames has been extensively studied over the past few years (Villegas et al., 2019; Denton & Fergus, 2018; Lee et al., 2018; Babaeizadeh et al., 2018; Vondrick et al., 2016; Finn et al., 2016; Clark et al., 2019; Castrejon et al., 2019). Early approaches focus on modeling simple and deterministic dynamics using regression loss (Srivastava et al., 2015; Ranzato et al., 2014) or predictive coding (Lotter et al., 2017). However, such deterministic models may not be appropriate for modeling stochastic variations in real-world videos. Recently, deep generative models have been employed to model dynamics in complex videos. Babaeizadeh et al. (2018) proposed a variational approach for modeling stochasticity in a sequence. Denton & Fergus (2018) incorporated more flexible frame-wise inference models. The prediction quality of the variational models has been further improved by employing rich structure in latent variables (Castrejon et al., 2019), maximally increasing the network parameters (Villegas et al., 2019), or incorporating adversarial loss (Vondrick et al., 2016; Clark et al., 2019; Tulyakov et al., 2018; Villegas et al., 2017a). Despite their success, these approaches still fail to generate long-term videos and the common artifacts were losing object structures, switching object categories, *etc*. Our approach addresses these issues by explicitly predicting dynamics in the low-dimensional label map using the hierarchical model.

Hierarchical models studied in the past for video prediction were usually in specific video domains (Villegas et al., 2017b; Wichers et al., 2018; Liang et al., 2017; Yan et al., 2018; Walker et al., 2017; Minderer et al., 2019). Villegas et al. (2017b) employed LSTMs to predict human body joints and visual analogy making to create textures, which is extended by Wichers et al. (2018) and Minderer et al. (2019) to unsupervised approaches. Other approaches employed sequential VAE (Yan et al., 2018) or GAN (Yang et al., 2018) to predict the human body posture. However, these approaches are designed specifically for certain objects, and evaluated under simple videos containing only a single moving object. Also, they mostly utilized deterministic models to learn structure-level dynamics. These simplified assumptions on video content and dynamics limit their application to real-world videos. We propose to resolve these limitations by using dense semantic label maps as universal representation and stochastic sequential estimator for modeling video dynamics.

## 4 EXPERIMENTS

### 4.1 EVALUATION METRICS

Below we describe the evaluation metrics used in the experiment. Unless otherwise specified, we use $64 \times 64$ images for all quantitative evaluation for fair comparison to the existing works.

**Short-term prediction** We employ three conventional metrics in the literature to evaluate the prediction performance on short-term videos (*i.e.,* less than 50 frames): VGG cosine similarity (CSIM) measuring the frame-wise perceptual similarity using VGG features, mean Intersection-over-Union (mIoU) measuring the structure-level similarity using the label maps extracted by the

pre-trained segmentation network (Zhu et al., 2019) and Fréchet Video Distance (FVD) (Unterthiner et al., 2018) measuring a Fréchet distance between the ground-truth videos and the generated ones in a video representation space. Detailed evaluation protocols are described in Appendix A.

**Long-term prediction** Compared to short-term prediction, evaluating prediction quality of arbitrary long-term video is challenging for a number of reasons. First, the ground-truth videos are seldom available at this scale, making it impossible to adopt metrics based on frame-wise comparison (*e.g.* CSIM and mIoU). Second, the uncertainty of future prediction increases exponentially with time, making it intractable to employ density-based metrics (*e.g.*, FVD) as it requires exponentially many samples. Since our goal is to demonstrate video prediction at the scale of hundreds to thousands of frames, we introduce a novel metric based on FVD, which enables evaluation of temporal and frame-level synthesis quality without ground-truth sequences and allows tractable evaluation through time. Specifically, we introduce a *shot-level* video-quality evaluation metric as

$$\text{shot-wise FVD}(t) = \text{FVD}(\hat{X}_{t:t+\omega-1}, X_\omega) \quad (7)$$

which computes a FVD between the ground-truth shots $X_\omega = \{X_\omega^1, \cdots, X_\omega^L\}$ and the shots of predictions $\hat{X}_{t:t+\omega-1} = \{\hat{X}_{t:t+\omega-1}^1, \cdots, \hat{X}_{t:t+\omega-1}^M\}$ in a sliding window manner through time, where $L$ denotes the total number of overlapping shots in the training video and $M$ denotes the number of predicted shots. The shot-wise FVD evaluates the synthesis quality in a short interval defined by $\omega$ *through time*. We show in the experiment that it is indeed aligned well with human perception. We also compute Inception Score (Salimans et al., 2016) for frame-level quality evaluation, which does not require ground-truths thus is appropriate for long-term evaluation. Please find Appendix A for more details.

**User study** We conduct an user study on Amazon Mechanical Turk (AMT). To evaluate the quality at different time scales, we present 5-second (50 frames) videos extracted at 1, 250, 400th predicted frames for all methods and counted their chosen ratio as the best. More details are in Appendix A.

## 4.2 RESULTS ON HUMAN DANCING SEQUENCES

We first present our results on human dancing videos collected from the internet (Wang et al., 2018).

**Baselines** We compare our method with two state-of-the-art video prediction models. SVG-extend (Villegas et al., 2019) is directly operating on RGB frames via a stochastic estimator and serves as our baseline for a non-hierarchical model. Following the paper, we employ the largest model with maximum parameters for fair comparison. We employ Villegas et al. (2017b) as a hierarchical model designed specifically for long-term prediction of human motion using pose (Newell et al., 2016). All models are trained to predict 40 future frames given 5 context frames.

**Short-term prediction results** We evaluate the short-term prediction performance by comparing the ground-truths with synthesized 50 future frames given 5 context frames. Table 1 summarizes the quantitative evaluation results (see Figure A for qualitative results). Even in a short prediction interval, our method generates substantially higher quality samples than the baselines in terms of modeling appearance (CSIM), structure (mIoU), and motion (FVD). This is because danc-

Table 1: Quantitative comparisons of short-term prediction results.

| Model | CSIM(↑) | mIoU(↑) | FVD(↓) |
|---|---|---|---|
| SVG-extend | 0.6654 | 0.0519 | 2125.29 |
| Villegas et al. | 0.7637 | 0.1755 | 1987.55 |
| Ours | **0.8164** | **0.3454** | **1398.98** |

ing sequences contain rapid and complex variations in dynamics, making prediction task particularly challenging. We discuss more detailed analysis across methods in a long-term prediction task below.

**Long-term prediction results** We evaluate 500 frame prediction results based on shot-wise FVD, frame-wise Inception score, and human evaluation. Figure 2 and 3 summarize the quantitative and qualitative comparisons, respectively. We notice that the SVG-extend fails to model complex dynamics in dancing even early in prediction. This is because the dancing motions involve fast transition and frequent self-occlusions, resulting in catastrophic error propagation through time. On the other hand, as shown in the Inception score, Villegas et al. (2017b) produces much reasonable future frames as it exploits human body joints for generation. However, the deterministic LSTM module is not strong enough to capture complex dancing motions, and ends up generating static postures in the long term that leads to very high FVD scores. In contrast, we observe that our method generates both realistic frames and convincing motions, leading to stable Inception and FVD scores in the long-term future. The human evaluation (Figure 2(c)) also shows the consistent results that our method outperforms all methods from the short- to long-term prediction. When our method is

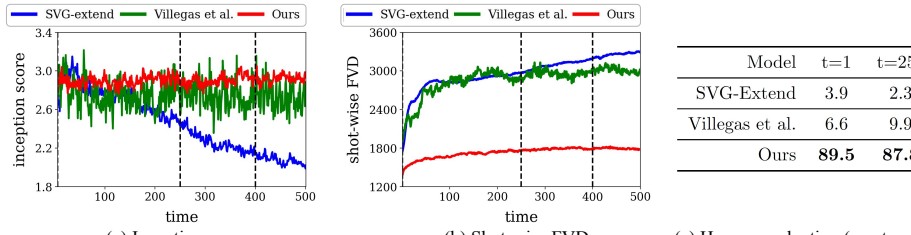

Figure 2: Quantitative comparisons of the long-term prediction on human dancing sequences.

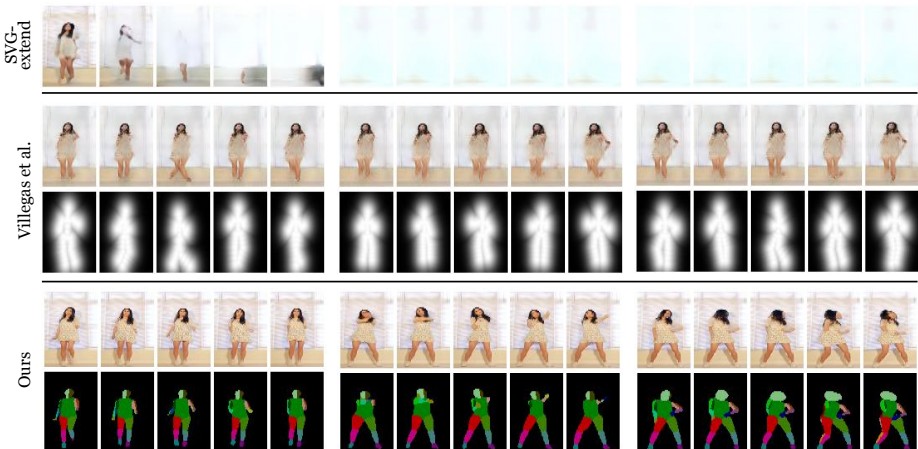

Figure 3: Qualitative comparisons of long-term video generation results across models on human dancing sequences. All models are conditioned on the same context frames. Click the image to play the video in a browser. More results are available at https://1konny.github.io/HVP/.

applied to predict up to 2040 frames in 128×128 resolution (see Figure C in the appendix), we observe that it generates future frames that are convincing and involve diverse dance motions without noticeable quality degradation through time. In addition to the synthesis quality, Figure B illustrates that our method can generate diverse and interesting motions with the stochastic structure estimator.

### 4.3   RESULTS ON KITTI BENCHMARK

**Baselines**   In addition to SVG-extend, we employ the future segmentation model (Bhattacharyya et al., 2019) (Bayes-WD-SL) that predicts future dense label maps as a strong baseline for hierarchical model. Since it generates only the label maps, we employ the same image generator with ours to produce RGB frames from the predicted label maps. All models are trained to predict 15 future frames given 5 context frames. Please refer to the appendix for the detailed architecture settings.

**Short-term prediction results**   We evaluate the short-term prediction quality by synthesizing 50 frames given 5 context frames. Table 2 summarizes the quantitative results (see Figure D for qualitative analysis). We observe that SVG-extend performs worse in structural accuracy (mIoU) and motion (FVD), as it loses the object structure and generates arbitrary pixel dynamics. The Bayes-WD-SL performs better in

Table 2: Quantitative comparisons of short-term prediction results.

| Model | CSIM(↑) | mIoU(↑) | FVD(↓) |
|---|---|---|---|
| SVG-extend | 0.6664 | 0.3529 | 1448.84 |
| Bayes-WD-SL | 0.6533 | 0.4225 | 956.05 |
| Ours | **0.6789** | **0.5137** | **762.73** |

mIoU, as it conditions the frame prediction with the structure estimation. However, as shown high FVD score, it fails to model temporal variations in the structure, largely due to its feedforward architecture. This unstructured motion leads to disastrous failure in a long-term prediction, as we will discuss later. In contrary, our method generates semantically accurate motions and structures, and substantially outperforms the others in all metrics.

**Long-term prediction results**   Figure 4 and 5 summarize the quantitative and qualitative comparisons, respectively. In all metrics, our method outperforms the other baselines with substantial margins, showing that our method synthesizes both high-quality frames and motion. We notice that SVG-extend simulates arbitrary pixel motions, resulting in relatively constant Inception and shot-wise FVD scores through time. However, these unstructured motions lead to rapid destruction in

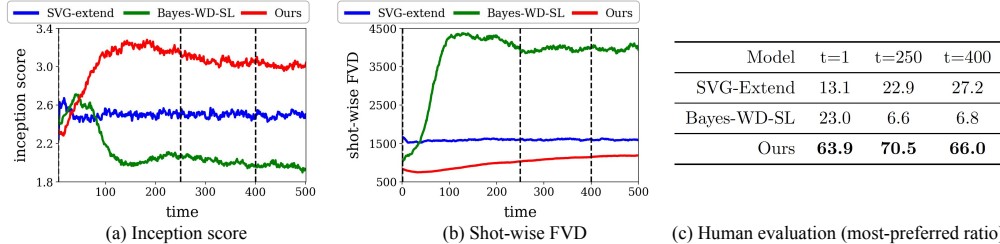

Figure 4: Quantitative comparisons of the long-term generation quality on KITTI Benchmark.

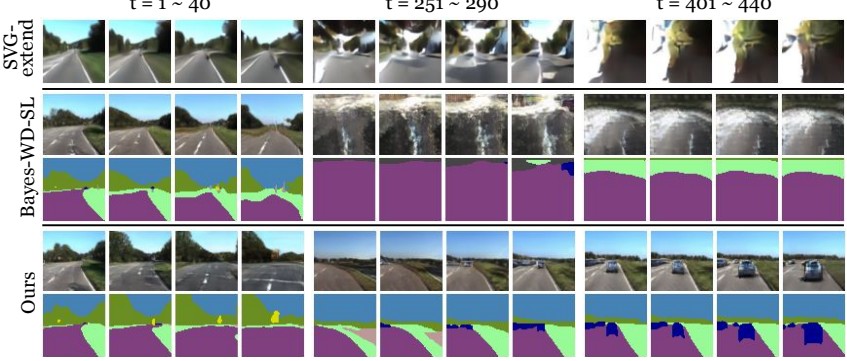

Figure 5: Qualitative comparisons of long-term video generation results across models. Although all models succeed in generating plausible frames in a short-term ($t = 1 \sim 40$), only our approach can generate persistent and convincing futures even in the end ($t = 251 \sim 500$). Click the image to play the video in a browser. More results are available at https://1konny.github.io/HVP/.

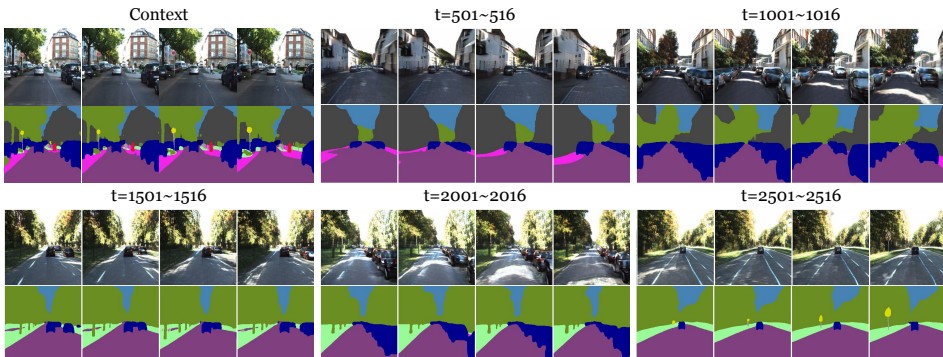

Figure 6: Long-term prediction results on a high-resolution KITTI sequence. Both the frames and the label maps are predicted by our method. Our method can generate high-resolution frames ($256 \times 256$ pixels) into the long-term future without particular quality degradation. Click the image to play the video in a browser. More results are available at https://1konny.github.io/HVP/.

recognizable concepts in the synthesized frames (Figure 5). The hierarchical model (Bayes-WD-SL) generates more convincing frames in a short-term as shown in lower shot-wise FVD, but fails to extrapolate in long-term. Importantly, such trends in shot-wise FVD (Figure 4(b)) aligns well with human evaluation results (Figure 4(c)), showing that it is appropriate metric for evaluation of long-term prediction. Interestingly, we observe that the Inception Score of our method increases through time. It is because the long-term prediction by our method often leads to scenes with simple and typical structures, such as a highway, where the image generator can produce more high-quality frames than complicated scenes appearing early in the test videos. Note that such transition of the scene is still reasonable as it is frequently observed in the training data. Nonetheless, we observe that our method generates reasonable sequences through time while maintaining its quality in a reasonable range even in a distant future. Finally, Figure 6 illustrates the prediction results over 2000 future frames. We observe that the generated sequences are reasonable in both structure and motion, and capture interesting translations of the scenes through time (*e.g.,* from suburban to rural areas).

Table 3: Comparison to future segmentation methods on Cityscapes dataset.

| Model | person | rider | car | truck | bus | train | motorcycle | bicycle | mIoU GT |
|---|---|---|---|---|---|---|---|---|---|
| S2S (Luc et al., 2017) | 0.37 | 0.18 | 0.70 | 0.43 | 0.55 | 0.26 | 0.27 | 0.38 | 0.39 |
| F2F (Luc et al., 2018) | 0.33 | 0.20 | 0.72 | 0.53 | 0.58 | 0.38 | 0.30 | 0.25 | 0.41 |
| Ours | **0.41** | **0.28** | **0.77** | **0.75** | **0.73** | **0.49** | **0.40** | **0.45** | **0.54** |

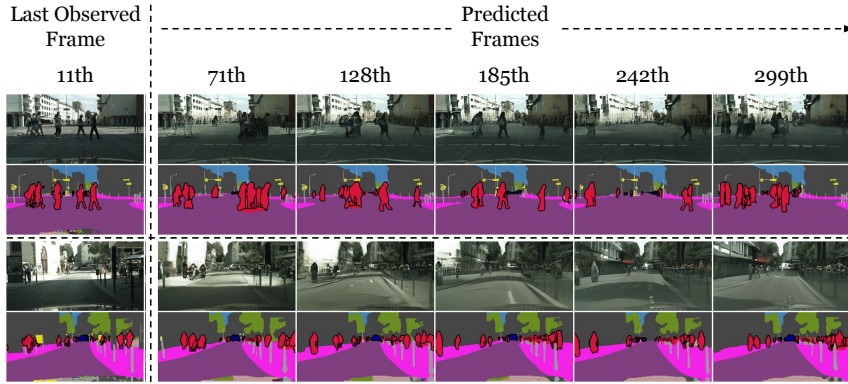

Figure 7: Long-term prediction results on $256 \times 512$ Cityscapes dataset. Click the image to play the video in a browser. For more qualitative results and details, please refer to Figure H and Figure J in the appendix and the project website: https://1konny.github.io/HVP/.

## 4.4 RESULTS ON CITYSCAPES DATASET

To further evaluate the quality of structure prediction, we compare our structure generator with existing future segmentation methods that directly predict the segmentation map of the future frames.

**Baselines** We compare our method with S2S (Luc et al., 2017) and F2F (Luc et al., 2018), which are the state-of-the-arts in the future segmentation literature. S2S is a deterministic model based on fully-convolutional network that predicts future semantic label maps in a multi-scale and auto-regressive manner. F2F is an extension of S2S to future *instance* segmentation task by predicting the high-level feature maps of Mask R-CNN (He et al., 2017). Similar to KITTI, we use the label maps extracted by the pre-trained semantic segmentation network (Zhu et al., 2019) for the training.

**Evaluation** We follow the standard evaluation protocols in the literature (Luc et al., 2017; 2018) for quantitative and qualitative comparisons. For each validation sequence, each model is provided with 4 contexts, and produces up to 29th frame. Then, we compute mIoUs between ground-truths and predictions at 20th time-step. For fair comparisons with the future instance segmentation model (*i.e.,* F2F), we follow Luc et al. (2018) and measure mIOUs only for moving objects. For S2S and F2F, we use publicly available pre-trained models provided by the authors.

**Results** Table 3 presents the quantitative evaluation results (See Figure H for qualitative results). The two baselines produce blurry predictions, resulting in inaccurate structures and mislabelings. These problems have been widely observed in the literature and are attributed to the inability to handle stochasticity (Denton & Fergus, 2018). Our method outperforms the two baselines in all classes as it can handle highly stochastic nature of complex driving scenes. Figure 7 and Figure J illustrates the long-term prediction results of our full model including the boundary map and image generator on $256 \times 512$ resolution. It shows that our method can generate convincing future even in extremely complex and high-resolution videos. See Appendix B.3 for more detailed discussion.

## 5 ABLATION STUDY

As demonstrated in the previous sections, the proposed hierarchical model succeeds at predicting frames *persistently* up to an arbitrary long-term future. To identify the source of this persistency, we conduct an ablation study. Specifically, we note that the model from Villegas et al. (2019), which suffers from dramatic error propagation through time, performs surprisingly well when applied to predicting semantic structures. Therefore, we focus on the different choices of structure generator and its hierarchical composition. We follow Villegas et al. (2019)and ablate effects of (1) stochastic estimation and (2) recurrent estimation. In addition, we investigate benefits of (3) discretization of predicted label maps and (4) our hierarchical model design that first predicts the structures and

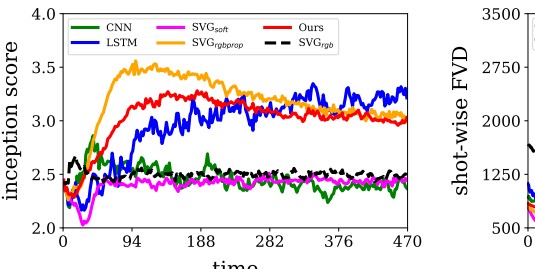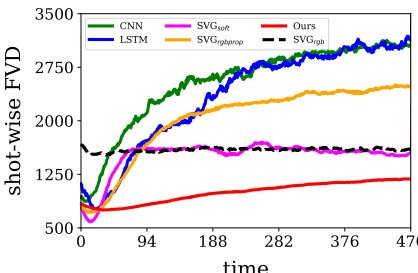

Figure 8: Quantitative comparisons of long-term prediction performances among ablated models.

motions via structure generator and then translates them into pixel-level RGB frames. The latter can naturally prevent the pixel-level errors from propagating through the structure predictions.

To this end, we construct four ablated baselines from our structure generator as follows: (1) `LSTM` where stochastic estimation modules are removed, (2) `CNN` where stochastic and recurrent estimation modules are removed, (3) $SVG_{soft}$ where the discretization process is removed and the model is modified to observe and estimate continuous logits of the semantic segmentation model, and (4) $SVG_{rgbprop}$ where an output of the structure generator at time $t$ is translated to RGB frame by the image generator and then further processed by the semantic segmentation model before used as an input at time $t + 1$. We train all the models in KITTI Benchmark on the $64 \times 64$ resolution and compare them using the inception score and shot-wise FVD. All baselines share the same image generator (Section 2.2) to translate the predicted structures to a RGB sequence. Figure 8 summarizes the quantitative results. For qualitative comparisons, please refer to Figure G in the appendix.

**Stochastic estimation.** As shown in the quantitative comparison results, deterministic baselines (`CNN` and `LSTM`) suffer from drastic degradation of shot-wise FVD through time compared to the stochastic ones. This is because the deterministic models tend to seek the most likely future and thus are prone to bad local minima, which are very difficult to be recovered from (*e.g.* when trapped in a loop and recursively generates the same structures and motions as shown in Figure G). It shows that the stochastic estimation is critical especially in a very long-term prediction.

**Recurrent estimation.** Thanks to the ability to capture temporal dependencies among time steps, we find that the recurrent model (`LSTM`) is much beneficial than the feedforward counterpart (`CNN`) in (1) predicting more plausible sequences in a short to mid-term (shot-wise FVD at $t < 200$) and (2) generating more recognizable scenes (inception score).

**Discretization.** We observe that predicting the structure in the *discrete* space is very critical in a persistent prediction. To validate this, we compare our method with the baselines (1) unrolling in RGB space ($SVG_{RGB}$) and (2) unrolling in structure but using soft class logits without discretization ($SVG_{soft}$). As shown in Figure G, both baselines perform much worse than our method. Surprisingly, the behaviors of both baselines are very similar in both measures, although they unroll in different spaces (RGB and soft class logit). It shows that unrolling in discrete space is one of the keys for persistent prediction, as it prevents error propagation and eliminates complex temporal variations except the ones caused by structure and motion.

**Decomposed hierarchical prediction.** Comparison to $SVG_{rgbprop}$ shows that errors induced from pixel-level predictions gradually add up through time and have a significant impact on predicting plausible structures and motions. It shows that our architecture having the independent unrolling loop of the structure and image generator is important for persistent prediction.

## 6 CONCLUSION

We proposed a hierarchical approach for persistent video prediction. We revisit the hierarchical model with stochastic estimation of dense label maps and integration of image generator robust against temporal mis-prediction in pixels and structures. Our experimental results show that a carefully designed hierarchical model can learn to synthesize a very long-term future with convincing structures and dynamics even in complex videos. By scaling up the video prediction to order-of-magnitudes longer, we believe that our work can help the future research to focus on more challenging problems of learning a long-term dependency and eliminating explicit structure estimation.

**Acknowledgment** This work was supported in part by Institute of Information & communications Technology Planning & Evaluation (IITP) grant funded by the Korea government (MSIT) (2020-0-00153 and 2016-0-00464), Samsung Electronics, and NSF CARRER under Grant 1453651.

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

## APPENDIX

## A EVALUATION PROTOCOLS

**Short-term prediction.** Following the previous works, we evaluate the short-term prediction quality by comparing the ground-truth and predicted frames in test videos. We employ three metrics: VGG cosine similarity (CSIM), mean Intersection-over-Union (mIoU), and Fréchet Video Distance (FVD) (Unterthiner et al., 2018). The first metric measures the frame-wise perceptual similarity using VGG features, while the second one measures the structure-level similarity using the label maps extracted by the pre-trained semantic segmentation network (Zhu et al., 2019). To handle stochasticity, we sample 100 predictions for each test clip and report the scores of the best scoring predictions. For the video-level evaluation, we adopt FVD that measures a Fréchet distance between the ground-truth videos and the generated ones in a video representation space. We use an Inflated 3D Convnet (I3D) (Carreira & Zisserman, 2017) network pre-trained on Kinetics 400 dataset (Carreira & Zisserman, 2017) to obtain representations on videos, and use all 100 predictions to compute FVD against ground-truth sequences.

**Long-term prediction.** To evaluate the shot-wise FVD, we employ the I3D Convnet as the short-term prediction. For the models trained on KITTI dataset, we use a ResNet-18 (He et al., 2016) pre-trained on Places 365 (Zhou et al., 2017) to compute the Inception Score. For the models trained on the Human Dancing dataset, we use the Inception V3 (Szegedy et al., 2015) pre-trained on ImageNet (Russakovsky et al., 2015) for measuring the Inception Score.

**Human evaluation.** We conducted a human evaluation using Amazon Mechanical Turk (AMT). For evaluating both the short- and long-term prediction performances, we generated 500 predicted frames for all compared methods (SVG-Extend, Villegas et al. (2019), and ours). We extracted 5-second clips (50 frames encoded in 10 fps) at the 1st, 250th, and 400th frames, and asked users to rank the videos according to perceived quality and realism of the videos.

On the KITTI dataset, we conducted the evaluation over the 133 unique sequences present in the validation set. On the Human Dancing dataset, we conducted evaluation over 97 videos used for validation. Responses for each video was aggregated from 5 independent human evaluators for each clip.

## B MORE DETAILS OF EXPERIMENT RESULTS

This section provides additional experiment details and results that could not be accommodated in the main paper due to space restriction. Please find our website to assess full videos for qualitative analysis: https://1konny.github.io/HVP/.

### B.1 HUMAN DANCING DATASET

#### B.1.1 DATASET

To evaluate if our model can learn complex and highly structured motion, we used videos of human dancing[2]. We construct this dataset by crawling a set of videos from the web containing a single person covering various dance moves. Following the pre-processing used in Wang et al. (2018), we crop the center of the video and resize it to square frames. We collect approximately 240 videos in total for training. For structure representation, we use the body parts segmentation obtained by DensePose (Alp Güler et al., 2018). We sub-sample each sequence by a factor of two.

#### B.1.2 IMPLEMENTATION DETAILS

**SVG-extend** (Villegas et al., 2019). We use the same architectural and hyperparameter settings described in Section C.1 except that SVG-extend in this case is trained to model pixel-level RGB

---

[2]https://www.instagram.com/imlisarhee/. We obtained the videos under the permission of the creator/owner.

sequences directly as an unsupervised non-hierarchical baseline, unlike ours that works on semantic segmentation sequences as sequential dense structure generator.

**Villegas et al. (2017b).** This method also adopts a hierarchical approach that leverages human keypoints as additional supervision for predicting future video frames. Specifically, this method consists of five modules, namely (1) the Hourglass network (Newell et al., 2016) that estimates human keypoints from pixel-level images, (2) a single layer deterministic encoder-decoder LSTM with 1024 hidden units and tanh activations that predicts a sequence of future human keypoints given a few observed ones, (3) an image generator that synthesizes pixel-level image frames given the predicted and observed keypoints based on visual-analogy scheme, (4) an image discriminator for adversarial learning, and (5) the AlexNet (Krizhevsky et al., 2012) pre-trained on ImageNet (Russakovsky et al., 2015) that is used for perceptual loss (Johnson et al., 2016). For training this model on our human dancing dataset, we directly follow the setting used in the paper (Villegas et al., 2017b), and use the publicly available code[3] provided by the authors.

### B.1.3    QUALITATIVE RESULTS

**Short-term prediction.**    We show in Figure A the qualitative comparisons of short-term predictions across models, which corresponds to the Table 1 in the main paper. As shown in the figure, SVG-extend starts to fail in producing plausible samples in a very short-term ($t = 8 \sim$), which can be attributed to its implicit modeling of spatio-temporal variations of structures and appearances. Villegas et al. (2017b) improves the quality of both structures and appearances as well as extrapolation through time via the explicit modeling of structures followed by the translation of predicted structures into pixel-level frames in a hierarchical manner. However, it still fails to produce realistic pixel-level frames due to the adoption of (1) coarse-grained human landmarks and deterministic model which are insufficient to fully describe the fine details of stochastic and structured motions, and (2) image synthesis performed by transforming the last observed frames into the future, which may not be appropriate when there exist large appearance changes due to self-occlusion and movement. On the other hand, our approach can produce high-fidelity predictions with plausible motions thanks to the stochastic and comprehensive estimation of scene dynamics under the semantic segmentation space.

**Diversity of samples.**    Figure B illustrates the multiple prediction results conditioned on the same context frames. We observe that our model produces diverse and plausible future frames, where such stochasticity mainly stems from the structure generator.

**Long-term prediction results.**    Figure C illustrates the prediction results on a very long-term future (over 2000 frames).We observe that our method generates convincing frames without particular quality degradation and error propagation over a very long-term future. We also observe that the predicted frames contain interesting dance motions, such as the ones with horizontal translation ($t = 1000 \sim 1040$), dynamics on legs ($t = 500 \sim 540$) and arms ($t = 2000 \sim 2040$), etc.

---

[3]https://github.com/rubenvillegas/icml2017hierchvid

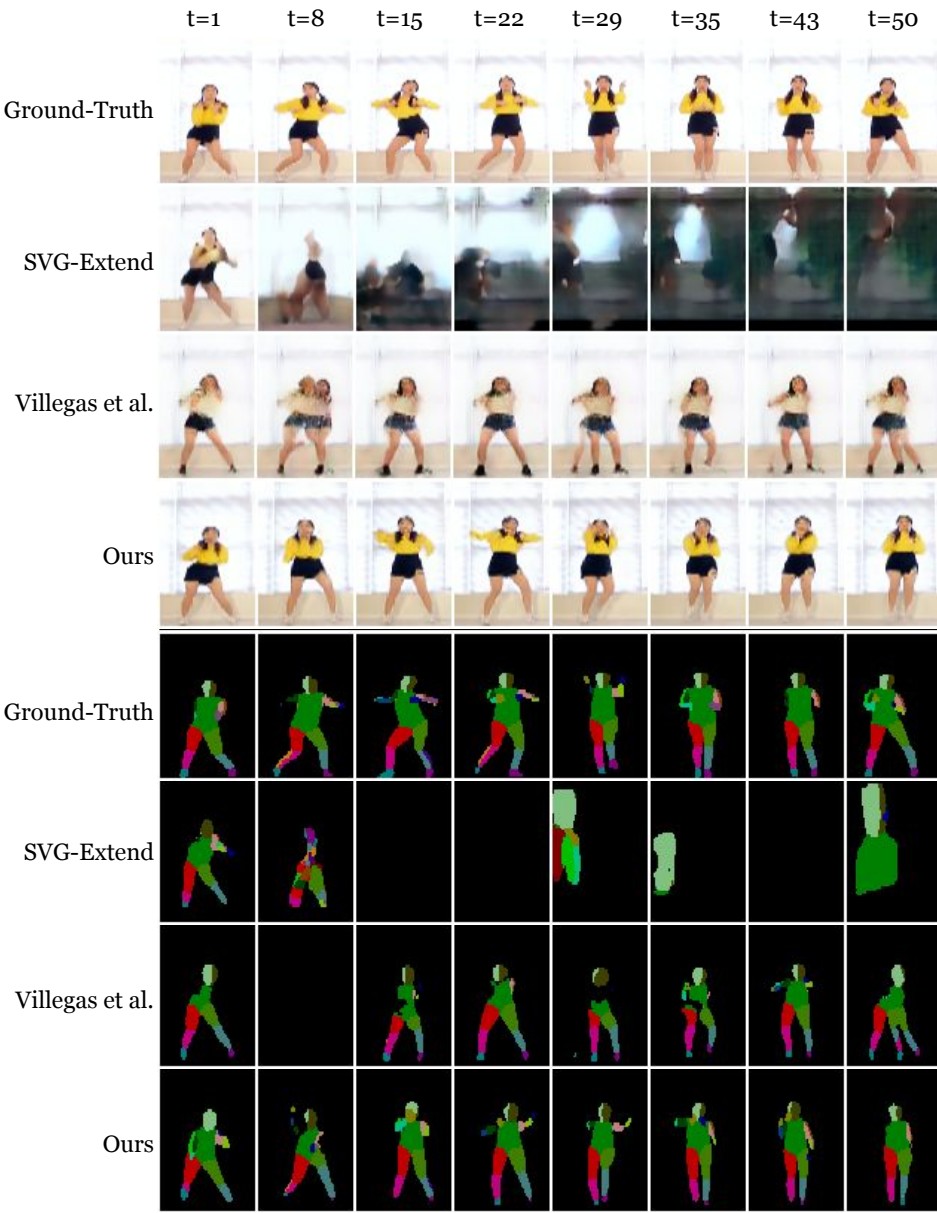

Figure A: Qualitative comparisons of frame-wise video prediction results across models: (top pane) given the same context frames, we sample 100 predictions for each model and show their best predictions in terms of the cosine similarity metric. (bottom pane) for each best prediction, we show the semantic label maps predicted by Densepose (Alp Güler et al., 2018).

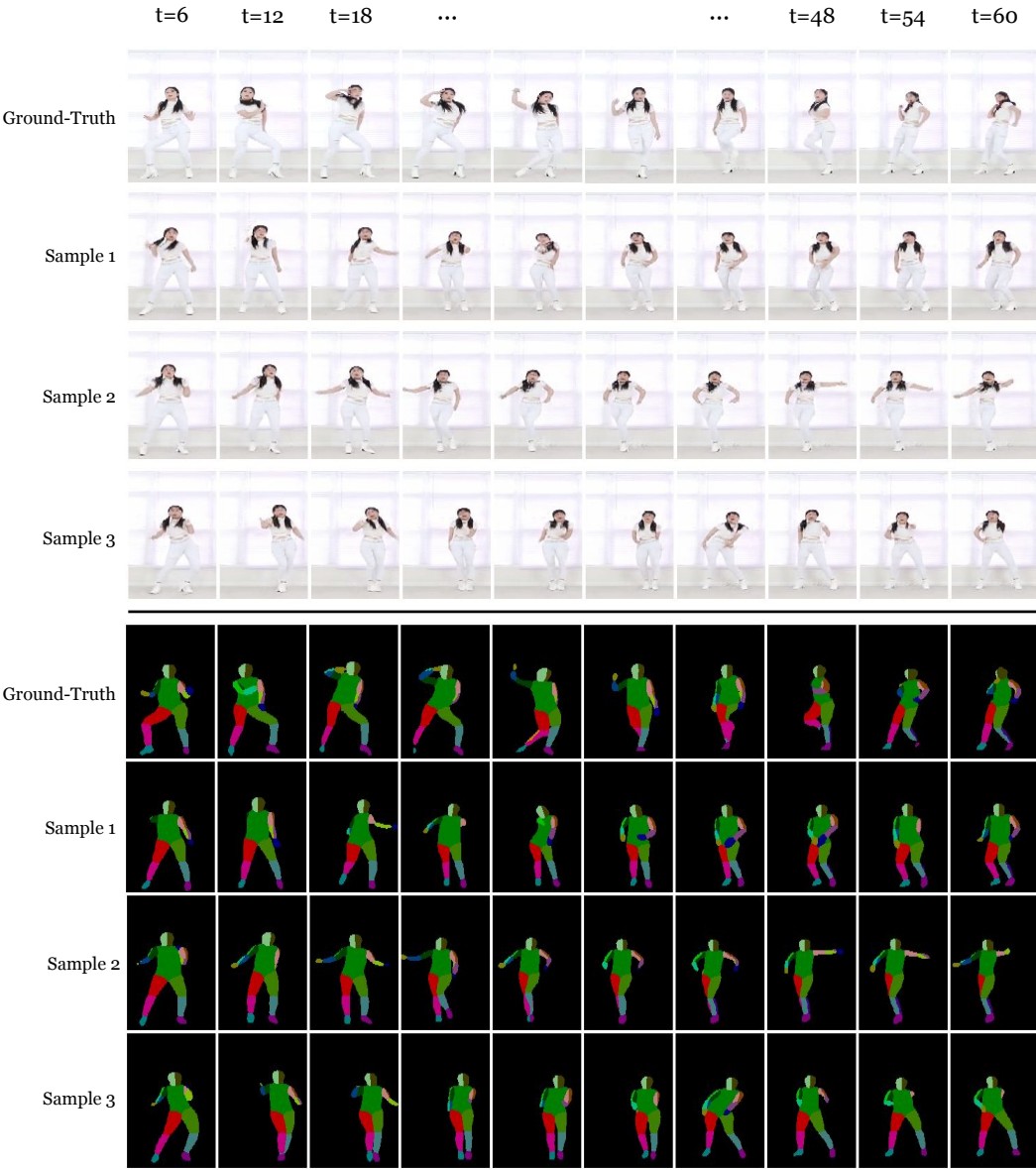

Figure B: Diverse video prediction results on a $128 \times 128$ dance sequence. All samples are conditioned on the same context frames.

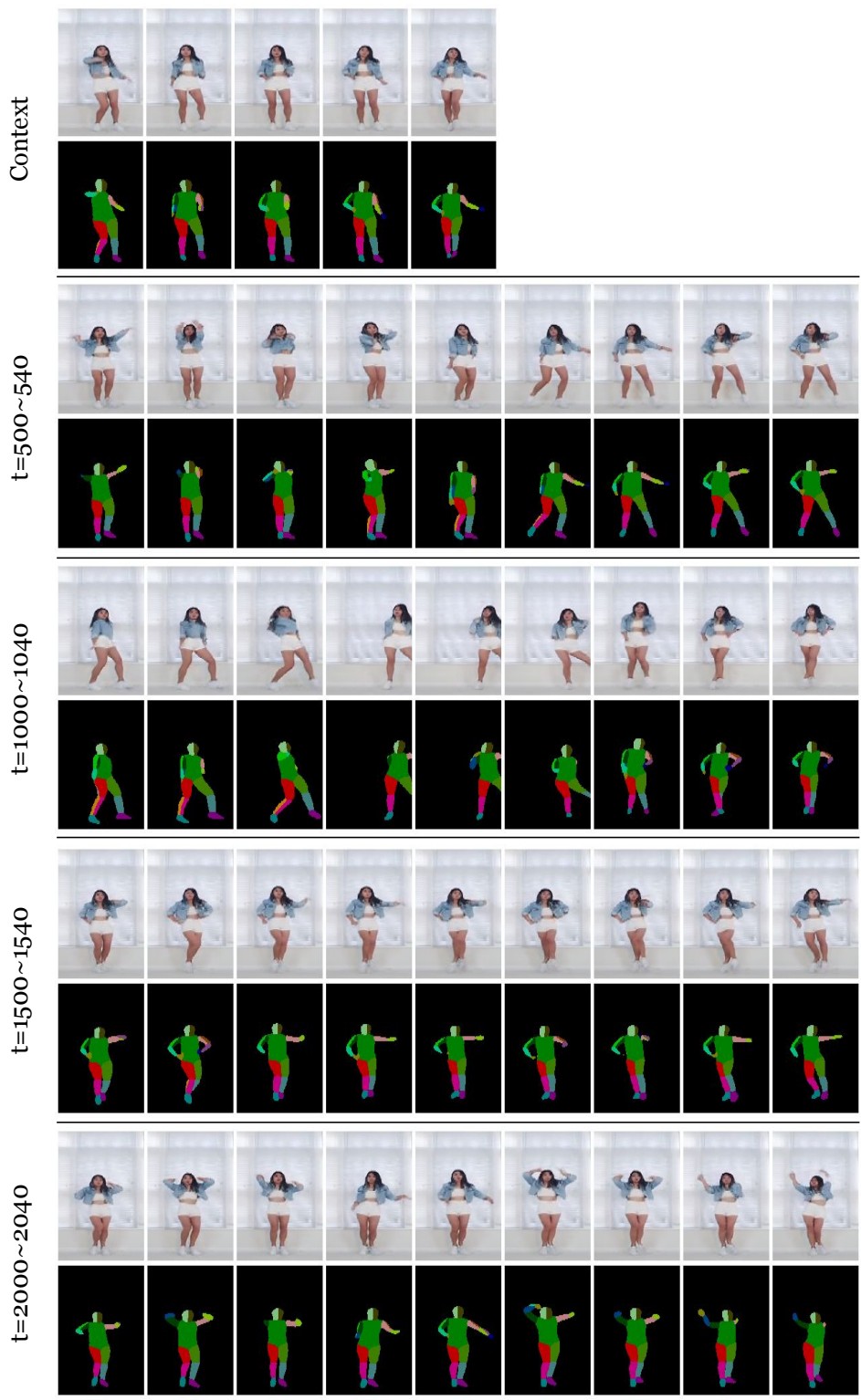

Figure C: Long-term prediction results on a $128 \times 128$ human dancing sequence. Please find our website to assess the full video.

## B.2 KITTI BENCHMARK

### B.2.1 DATASET

An ability to cope with partial observability and high stochasticity is highly important in the video prediction task. To evaluate a model under this criterion, we adopt KITTI dataset (Geiger et al., 2013) consisting of driving scenes captured by a front-view camera in residential and rural areas. Since the segmentation labels are available for only a small subset of frames in this dataset (200 frames in total), we extract the labels using the semantic segmentation model (Zhu et al., 2019) pre-trained on still images (Cordts et al., 2016; Geiger et al., 2013) for both training (the entire sequences) and testing (only context frames). We follow the pre-processing and train/test splits used in Lotter et al. (2017), and sub-sample each sequence by a factor of two.

### B.2.2 IMPLEMENTATION DETAILS

**SVG-extend** (Villegas et al., 2019). Same as Section B.1.2, we follow the implementation details specified in Section C.1 for training this model on KITTI Benchmark dataset.

**Bayes-WD-SL** (Bhattacharyya et al., 2019). This work is a future segmentation method that exploits odometry information as additional supervision and enables stochastic estimation of uncertain futures through the Bayesian approach. On top of a ResNet-based deterministic video prediction model (Luc et al., 2017), this work introduces a novel Bayesian formulation to better handle inherent uncertainties when predicting future frames. Specifically, this method introduces (1) weight-dropout (WD) which assumes a finite set of weights for approximating model space, followed by Bernoulli variational distribution on those finite set of weights, and (2) synthetic likelihoods (SL) which mitigates the constraint on each model to explain all the data points and thus allows sampled models to be diverse to better deal with the uncertainties. For training this model on KITTI Benchmark, we follow the same settings used in the original paper for the experiments on Cityscapes dataset except that we omit odometry supervision for fair comparisons. We use the publicly available code.[3]

### B.2.3 QUALITATIVE RESULTS

**Short-term prediction.** Figure D illustrates the qualitative comparisons for short-term prediction, which correspond to Table 2 in the main paper. As shown in Figure D, SVG-extend performs very well at the very beginning of the prediction ($t = 1 \sim 8$), since it directly exploits the content from the last context frame via temporal skip-connection. However, its prediction quality degrades rapidly through time, as it loses the object structure and simulates arbitrary, unstructured pixel-wise motions in the long-term. The future segmentation baseline (Bayes-WD-SL) preserves a much reasonable structure, since its generation is conditioned on the estimated label maps similar to ours. However, it fails to model structures in *motion* and thus loses such structures in the end. These results demonstrate that both the unsupervised and hierarchical baselines fail to *extrapolate* to the long-term horizon than the one used during the training. Compared to these baselines, our method generates semantically more accurate motions and structures, and substantially outperforms the others especially in the latter prediction steps.

**Diversity of samples.** We show diverse prediction results in Figure E, which are conditioned on the same context frames. As can be seen in the figure, predicted sequences not only are accurate in a near future ($t = 18$), but also show diverse motions and scene dynamics through time. For example, we can see diverse scene dynamics such as emergence of novel vehicles ($t = 54 \sim 144$), as well as transition to the novel scene ($t = 144 \sim 180$).

**Long-term prediction results.** Figure F illustrates the long-term prediction results over 2000 frames in $256 \times 256$ resolution, which corresponds to Figure 6 in the main paper. We observe that the generated sequences are reasonable in both structure and motion. Interestingly, unrolling the prediction over a very long-term leads to very interesting transitions of the scene, such as translation from suburban ($t < 1035$) to rural areas ($t > 1500$), which cannot be captured in a short-term period. We believe that these results illustrate well why the long-term prediction is important.

---

[3]https://github.com/apratimbhattacharyya18/seg_pred

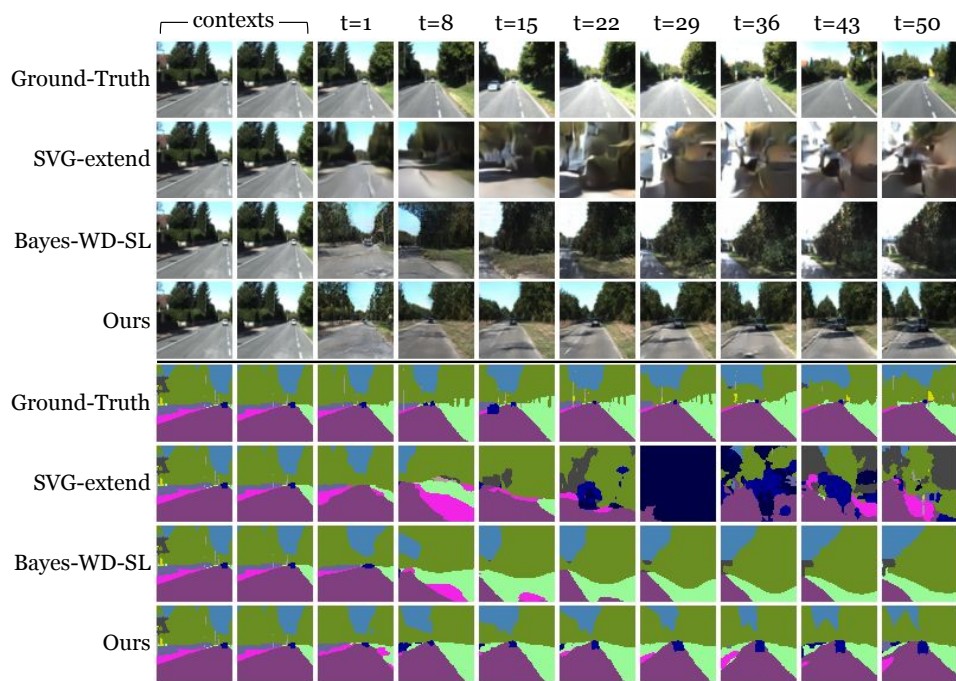

Figure D: Qualitative comparisons of frame-wise video prediction results across models: (top pane) given the same context frames, we sample 100 predictions for each model and show their best predictions in terms of the cosine similarity metric. (bottom pane) for each best prediction, we show the semantic label maps predicted by pre-trained semantic segmentation network (Zhu et al., 2019).

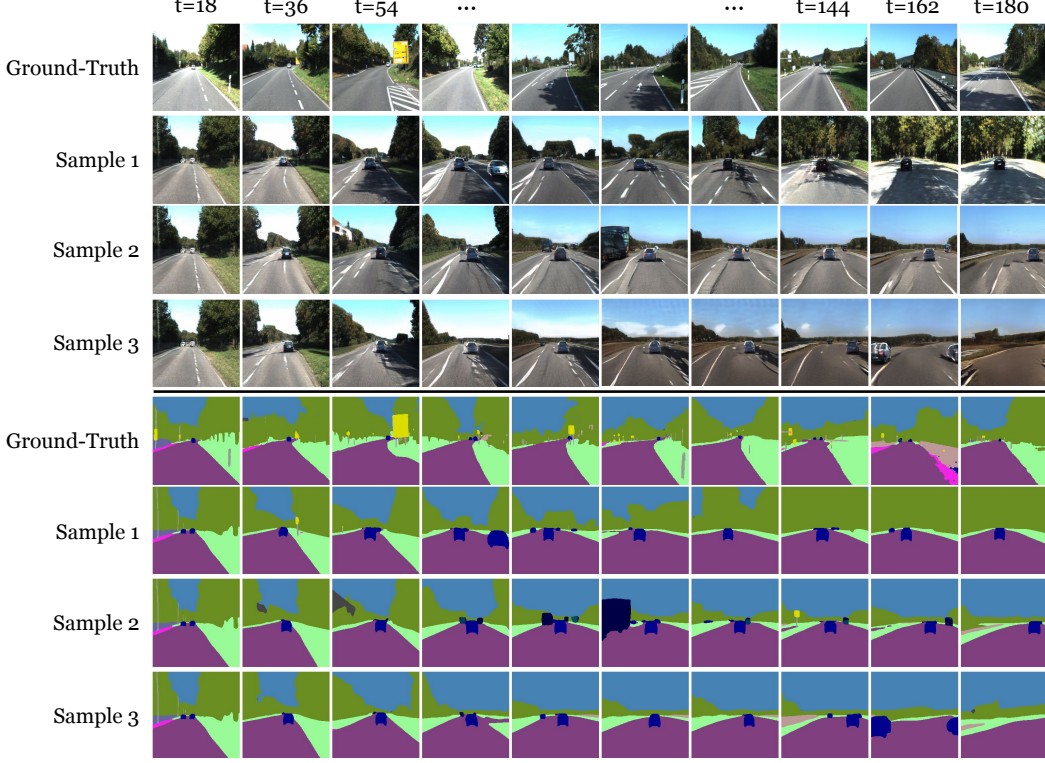

Figure E: Diverse video prediction results on a $256 \times 256$ KITTI sequence. All samples are conditioned on the same context frames.

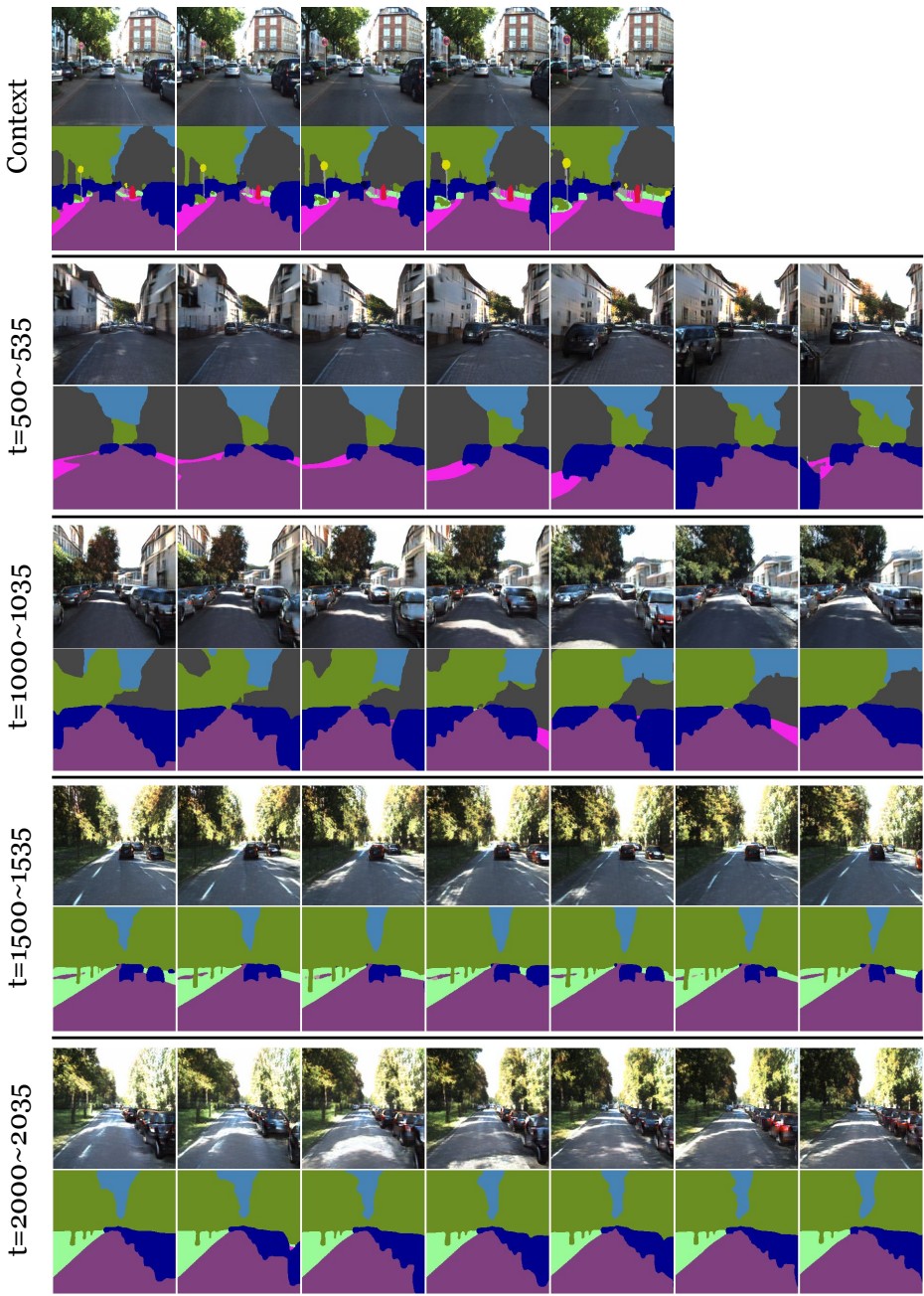

Figure F: Long-term prediction results on a high-resolution KITTI sequence. Both the frames and the label maps are predicted by our method. Our method can generate high-resolution frames ($256 \times 256$ pixels) into the long-term future without particular quality degradation.

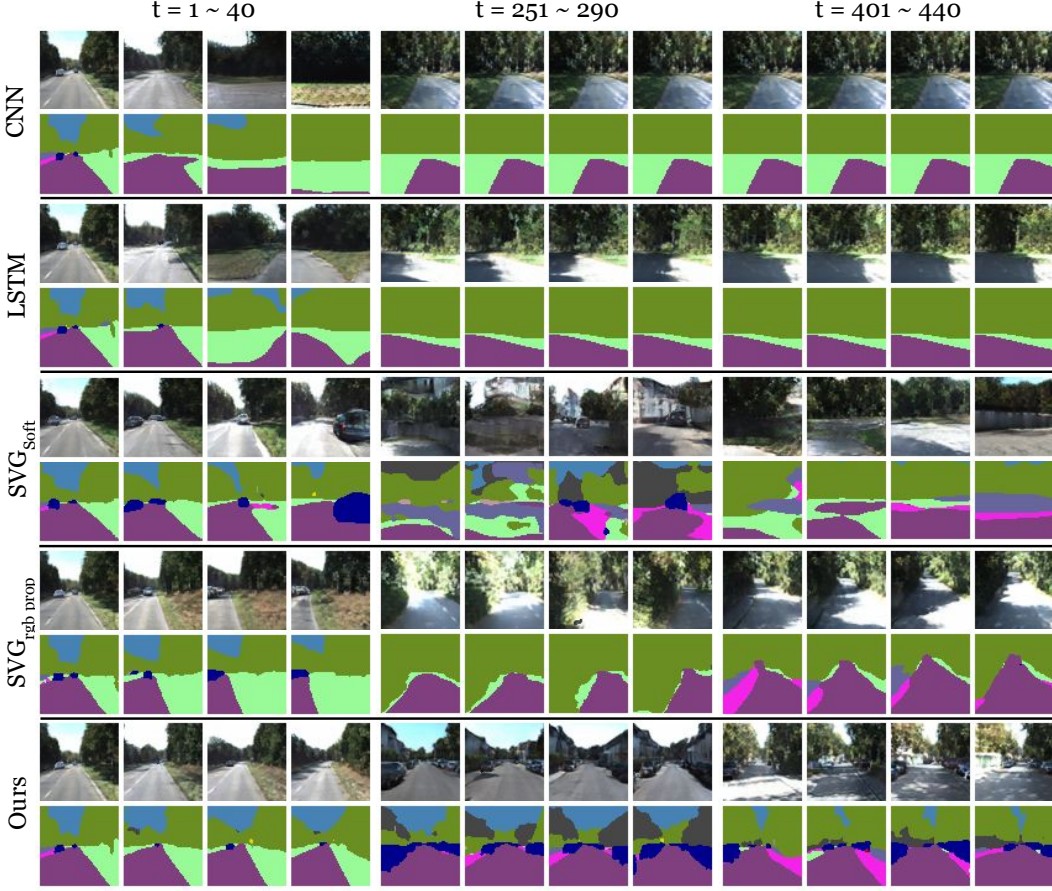

Figure G: Qualitative comparisons of long-term generation results across the ablated models. We ablate SVG-extend to produce four baselines: `LSTM` where stochastic estimation modules are removed, `CNN` where stochastic and recurrent modules are removed, $SVG_{soft}$ where the discretization process is removed and the model is modified to observe and estimate continuous logits of semantic segmentation model, and $SVG_{rgbprop}$ where an output of the structure generator at time $t$ is translated to RGB frame by the pixel generator and then further processed by the semantic segmentation model before used as an input at time $t + 1$. Please refer to the Section 5 for the discussion.

### B.3 CITYSCAPES DATASET

#### B.3.1 DATASET

To evaluate the quality of forecasting dense labels by the structure generator, we employ the Cityscapes (Cordts et al., 2016), a widely-used benchmark in future segmentation tasks. The dataset consists of 2,975 training, 1,525 testing, and 500 validation sequences where each sequence is 30-frame-long and has ground-truth segmentation labels only in the 20th frame. Similar to KITTI dataset, we employ the pre-trained segmentation network (Zhu et al., 2019) to generate segmentation labels of $256 \times 512$ image resolution for training our SVG-extend. Following (Luc et al., 2017; 2018), we subsample frames by a factor of three for each sequence and test all models on $128 \times 256$ image resolution.

#### B.3.2 IMPLEMENTATION DETAILS

**S2S** (Luc et al., 2017). We use the best performing S2S model proposed by the authors, namely *S2S-dil, AR, fine-tune*, which improves the performance of the single-frame prediction baseline (S2S) by introducing the followings: (1) dilated convolution to enlarge receptive fields of the model. (2) autoregressive fine-tuning that uses the prediction at time $t$ as an input to predict the future frame at time $t + 1$ during training, which enables the single-frame prediction model to predict an arbitrary number of frames into the future. We use a publicly available pre-trained model[4] provided by the authors, which was trained using the predicted semantic segmentation maps by Dilation10 network (Yu & Koltun, 2016) on $128 \times 256$ image resolution.

**F2F** (Luc et al., 2018). This method targets a future instance segmentation task, which extends future segmentation method S2S (Luc et al., 2017) by formulating the task of the model as predicting intermediate feature maps of Mask R-CNN (He et al., 2017) on the target future instance segmentation masks. We use a publicly available pre-trained model and code[5] provided by the authors, which was first trained on MS-COCO dataset (Lin et al., 2014) and then finetuned on the Cityscapes images of $1024 \times 2048$. Following (Luc et al., 2017; 2018), we down-sample the predicted future instance segmentation masks to $128 \times 256$ resolution for the evaluation.

#### B.3.3 QUALITATIVE RESULTS

**Qualitative comparison.** We present the qualitative comparisons of our method to existing future segmentation models, which correspond to Table 3 in the main paper. As shown in the table, our method outperforms the two baselines in all classes, thanks to its ability to handle the highly stochastic nature of complex driving scenes. We also compare qualitative samples of each model in Figure H. As shown in the figure, our model predicts convincing structures and their temporal dynamics are accurate. On the other hand, the two baselines produce blurry predictions, resulting in (1) inaccurate structures (S2S at all timestamps and F2F at 17-29th timestamps), and (2) multiple semantics in a single structure (S2S at all timestamps and F2F at 23-29th timestamps). These blurred prediction problems have been widely observed in the literature and attributed to the inability of deterministic models to handle stochasticity in data (Denton & Fergus, 2018).

**Short-term prediction.** To evaluate the future segmentation results in Section 4.4, we extract 6 future semantic or instance segmentations ahead (14, 17, 20, 23, 26, and 29th) from each model given 4 contexts (2, 5, 8, and 11th). Then, following (Luc et al., 2018), we compute per-class and mean IoUs for 8 moving object categories only: *person, rider, car, truck bus, train, motorcycle, bicycle*.

**Long-term video prediction.** Figure J illustrates the qualitative results of video prediction results on the Cityscapes dataset. As the Cityscapes contains many occluding objects, we employ the extended structure generator with instance boundary prediction to improve the frame prediction quality. We show in Figure J the predicted sequences up to 96 future frames conditioned on 4 context frames. As shown in the figure, our method can easily incorporate the additional structures (*i.e.* instance boundary map), and can be generalized to complex and high-resolution sequences

---

[4]https://github.com/facebookresearch/SegmPred
[5]https://github.com/facebookresearch/instpred

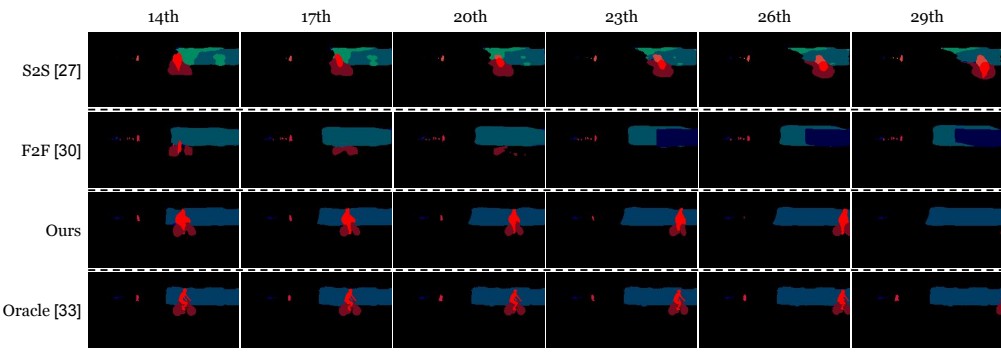

Figure H: Quantitative comparisons of future segmentation models. All models predict structures of moving objects (8 classes) up to 29th frame given 4 context frames (2, 5, 8, and 11th). The oracle is obtained by running the pre-trained segmentation model on the ground-truth frames. The results of compared methods (S2S and F2F) are based on the pre-trained models provided by the authors.

to predict instance-wise structures into the long future without error propagations (*e.g.* blurred predictions in the structure space). On the other hand, we also observe some undesirable biases in the predicted motion dynamics in the structure space, such as static or cyclic motions of the structures (e.g.red dashed boxes in Figure J). We notice that it is because the training sequences in Cityscapes datasets are considerably short (*i.e.* 30 frames), which makes the structure generator favor short-term dependencies in the prediction (*e.g.* short-term skip connections). It leads to biases in the model and artifacts in long-term extrapolation, such as static motions in the distant objects (*e.g.* buildings and trees far away from the camera), fixed content (*e.g.* repeatedly generated objects), and *etc.*, which are not observed in the other datasets with reasonably long training sequences.

### B.3.4 THE EFFECT OF INSTANCE BOUNDARY MAP PREDICTION

To analyze the effect of the instance boundary map produced by $G_{\text{edge}}$, we compare the prediction performance of our model with and without $G_{edge}$. Specifically, we extract 6 future frames ahead (14, 17, 20, 23, 26, and 29-th) given 4 context frames (1, 4, 7, and 11-th) from the models, and compare frame-wise evaluation results measured by Peak Signal-to-noise ratio (PSNR), Struc-

Table A: Quantitative results of ablating $G_{\text{edge}}$ on the $256 \times 512$ Cityscapes dataset.

| Model | PSNR($\uparrow$) | SSIM($\uparrow$) | CSIM($\uparrow$) |
|---|---|---|---|
| With $G_{\text{edge}}$ | 20.427 | 0.610 | 0.938 |
| Without $G_{\text{edge}}$ | 19.236 | 0.602 | 0.937 |

tural Similarity (SSIM), and VGG cosine similarity (CSIM). We average the measured scores across the test sequences and timestamps. Table A and Figure I show the quantitative and qualitative results, respectively. As shown in the table, we observe marginal improvements in those metrics due to the relatively short length of Cityscapes sequences (up to 30 frames). However, as can be seen in the figure, we find that the generated frames contain more clear instance boundaries with $G_{edge}$ especially in crowded scenes.

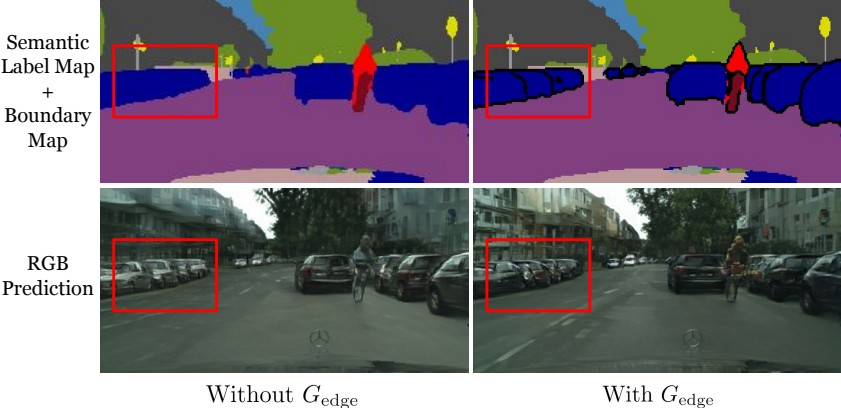

Without $G_{\text{edge}}$         With $G_{\text{edge}}$

Figure I: Qualitative results of ablating $G_{\text{edge}}$ on the $256 \times 512$ Cityscapes dataset.

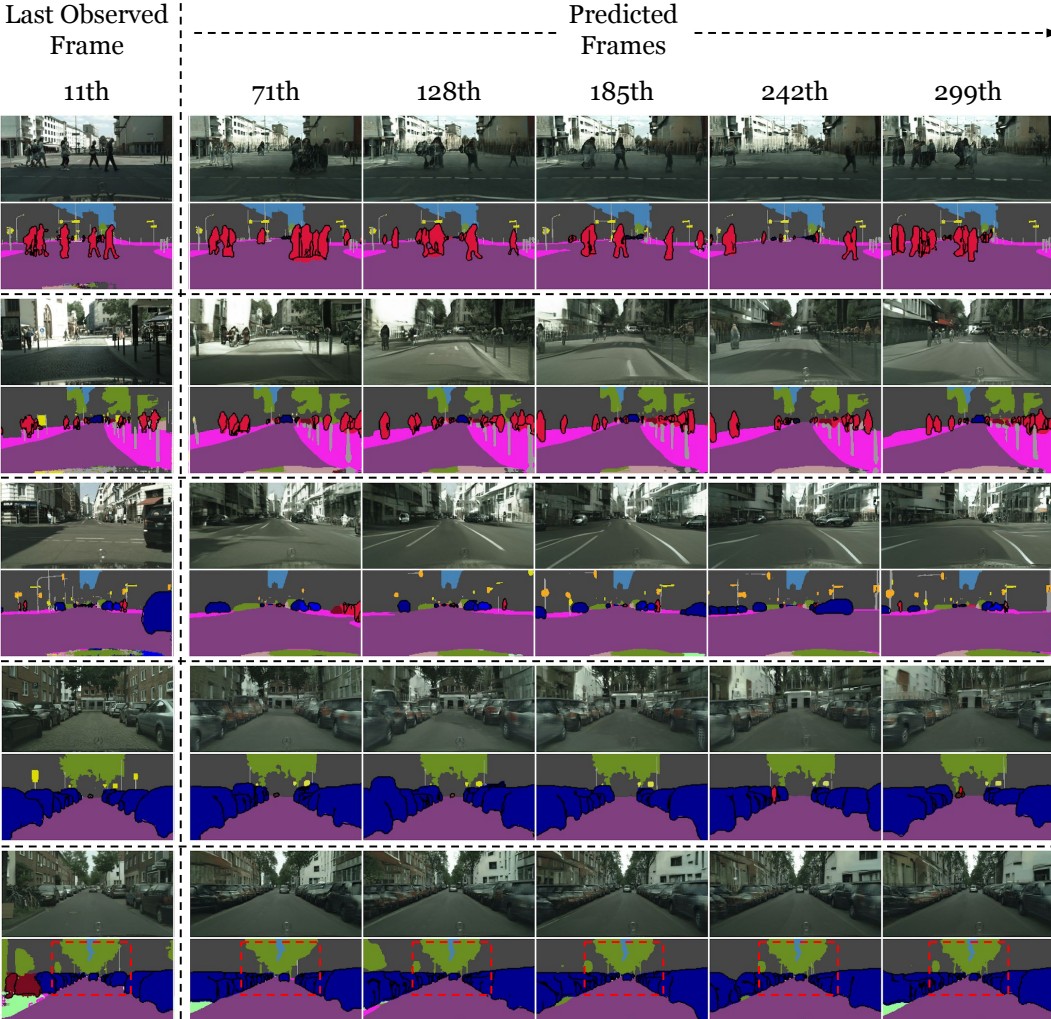

Figure J: Long-term prediction results on $256 \times 512$ Cityscapes dataset. We predict 96 future frames $(14, 17, \cdots, 299\text{th})$ given 4 context frames. Our structure generator is extended to produce additional instance boundary map for improved instance boundary demarcation in the image generator. Red dashed boxes indicate some biases in the prediction, such as static and cyclic motions in the predicted structures, which are attributed to the short training sequences in the Cityscapes dataset. Please refer to Section 4.4 for the description.

## C   MODEL ARCHITECTURE

In this section, we detail the architectures and hyperparameters used in our experiments.

### C.1   SEQUENTIAL DENSE STRUCTURE GENERATOR

We use the modified version of SVG (Denton & Fergus, 2018) by Villegas et al. (2019) as our sequential dense structure generator, which we refer to as SVG-extend throughout the paper. The two main differences between SVG and SVG-extend are two-fold: (1) SVG-extend adopts the shallower auto-encoder architecture than that of SVG. (2) SVG-extend replaces Fully-Connected-LSTM with Convolutional-LSTM (Shi et al., 2015). These two modifications are to better exploit spatial information contained in 3D feature maps, which is not maintained when inputs are mapped to low-dimensional 1-D image embedding vectors.

Since the detailed architectural configuration of SVG-extend is not described in (Villegas et al., 2019), we summarize our reproduced version of this framework in Table B, C, D, and E. On the $64 \times 64$ image resolution, we use the same architecture described in the tables. On $128 \times 128$ and $256 \times 256$ image resolutions, we simply stack one and two more convolutional blocks (non-linear convolutional layers followed by upsampling or downsampling layers) in the Encoder and Decoder networks, respectively, so that the shape of $\mathbf{h}_t, \mathbf{z}_t, \mathbf{g}_t$ is kept to $\mathbb{R}^{128 \times 8 \times 8}$ agnostic to the image resolutions. For hyperparameters, we use $C = 5$; we set $\beta = 0.0005$; we use ADAM optimizer (Kingma & Ba, 2015) with the learning rate of $0.0001$ and $(\beta_1, \beta_2) = (0.9, 0.999)$.

### C.2   BOUNARY MAP PREDICTION

When the pre-trained instance-wise segmentation model is available, we can optionally extend the structure generator to predict the object boundary maps as well. Such structures can add a notion of object instance and is useful to improve the image generator in sequences with many occluding objects. Let $\mathbf{e}_t \in \{0, 1\}^{H \times W}$ denotes the object boundary map at frame $\mathbf{x}_t$. Then we train the denoising autoencoder $G_{\text{edge}}$ to produce an edge map of each frame by

$$\hat{\mathbf{e}}_t = G_{\text{edge}}(\hat{\mathbf{s}}_t). \tag{8}$$

We employed the encoder-decoder network (Nazeri et al., 2019) as the conditional generator, and train it based on adversarial loss by

$$\mathcal{L}_{adv} = \mathbb{E}_{\mathbf{e},\mathbf{s}}[\log D(\mathbf{e}, \mathbf{s})] + \mathbb{E}_{\mathbf{s}}[\log(1 - D(\mathbf{e}, G_{\text{edge}}(\mathbf{s})))] \tag{9}$$

where $D$ is the discriminator. Note that the edge generator in Eq. 8 is applied to each frame independently; we observe that adding the predicted boundary map as an input to the structure generator makes the model be prune to error propagation. The boundary and label maps are then combined as the output of the structure generator $\tilde{\mathbf{s}}_t = [\hat{\mathbf{s}}_t, \hat{\mathbf{e}}_t]$.

### C.3   STRUCTURE-CONDITIONAL PIXEL SEQUENCE GENERATOR

We use Vid2Vid (Wang et al., 2018) for our structure-conditional pixel sequence generator. For Vid2Vid on KITTI dataset of $64 \times 64$ resolution, we reduce the number of downsampling layers in the generator and the scale parameter of the patch discriminator to 1. We use $\tau = 5$ for KITTI and Human Dancing and use $\tau = 4$ for Cityscapes. We use $\tau' = 3$ for all experiments. If not specified otherwise, we do not modify any hyperparameters and simply follow the default settings for the training.

Table B: Encoder Architecture.

| Input: $\mathbf{s}_t \in \mathbb{R}^{N \times 64 \times 64}$ |
| --- |
| $3 \times 3$ Conv 64, ReLU |
| $3 \times 3$ Conv 64, ReLU $\to \mathbf{h}_t^1$ |
| $2 \times 2$ MaxPool |
| $3 \times 3$ Conv 128, ReLU |
| $3 \times 3$ Conv 128, ReLU $\to \mathbf{h}_t^2$ |
| $2 \times 2$ MaxPool |
| $3 \times 3$ Conv 256, ReLU |
| $3 \times 3$ Conv 256, ReLU |
| $3 \times 3$ Conv 256, ReLU $\to \mathbf{h}_t^3$ |
| $2 \times 2$ MaxPool |
| $3 \times 3$ Conv 128, ReLU $\to \mathbf{h}_t$ |

Table C: Posterior/Prior LSTM Architecture.

| Input: $\mathbf{h}_t \in \mathbb{R}^{128 \times 8 \times 8}$ |
| --- |
| $3 \times 3$ ConvLSTM 256, $3 \times 3$ Conv 256, ReLU $\to \mu(t), \sigma(t), \mathbf{z}_t \sim \mathcal{N}(\mu(t), \sigma(t)) \in \mathbb{R}^{128 \times 8 \times 8}$ |

Table D: Predictor LSTM Architecture.

| Input: $[\mathbf{h}_t, \mathbf{z}_t] \in \mathbb{R}^{(128+128) \times 8 \times 8}$ |
| --- |
| $3 \times 3$ ConvLSTM 256, ReLU |
| $3 \times 3$ ConvLSTM 256, ReLU |
| $3 \times 3$ Conv 128, ReLU $\to \mathbf{g}_t \in \mathbb{R}^{128 \times 8 \times 8}$ |

Table E: Decoder Architecture.

| Input: $\mathbf{g}_t \in \mathbb{R}^{128 \times 8 \times 8}$ |
| --- |
| $2 \times 2$ NN Upsample $\to \mathbf{g}_t^3$ |
| Concatenate($\mathbf{g}_t^3, \mathbf{h}_t^3$) |
| $3 \times 3$ Conv 256, ReLU |
| $3 \times 3$ Conv 256, ReLU |
| $3 \times 3$ Conv 128, ReLU |
| $2 \times 2$ NN Upsample $\to \mathbf{g}_t^2$ |
| Concatenate($\mathbf{g}_t^2, \mathbf{h}_t^2$) |
| $3 \times 3$ Conv 128, ReLU |
| $3 \times 3$ Conv 64, ReLU |
| $2 \times 2$ NN Upsample $\to \mathbf{g}_t^1$ |
| Concatenate($\mathbf{g}_t^1, \mathbf{h}_t^1$) |
| $3 \times 3$ Conv 64, ReLU |
| $3 \times 3$ Conv 3, LogSoftmax $\to \hat{\mathbf{s}}_t$ |

Table F: Time required to train the model.

| Dataset | Model | # GPU (V100 16GB) | Training Time |
|---------|-------|-------------------|---------------|
| KITTI & Human Dancing $64 \times 64$ | SVG-Extend | 1 | 9 hours |
| | Vid2Vid | 1 | 12 hours |
| Human Dancing $128 \times 128$ | SVG-Extend | 4 | 48 hours |
| | Vid2Vid | 4 | 48 hours |
| KITTI $256 \times 256$ | SVG-Extend | 8 | 48 hours |
| | Vid2Vid | 8 | 48 hours |
| CittyScapes $256 \times 512$ | SVG-Extend | 8 | 48 hours |
| | Edge Predictor | 1 | 12 hours |
| | Vid2Vid | 8 | 48 hours |

