# OpenReview forum: "Revisiting Hierarchical Approach for Persistent Long-Term Video Prediction"
_ICLR.cc/2021/Conference — ICLR 2021 Poster_

### Official Review · AnonReviewer1 · 2020-10-23
**Review 1: reasonable paper with good results**

**Rating:** 6
**Confidence:** 4

**Review:**

---- Summary ----

The paper extends video-to-video translation model of (Wang’18) to video prediction by first generating a sequence of segmentation masks and then translating them into videos. Variational video prediction is used to generate a sequence of segmentation masks. The model produces impressive high-resolution and long-horizon results, and is extensively evaluated on Kitti, Cityscapes, and dancing data, outperforming some previously proposed methods.

---- Decision ----

The paper proposes a relevant method for hierarchical video prediction with impressive high-resolution and long-horizon results. As the paper notes, this sets a new standard for video prediction methods, and will likely spur more research into achieving similar results without the labeled data requirement. I am willing to accept the paper provided the author’s response clarifies my questions.

---- Strengths ----

The paper presents a modern version of Villegas’17a, powered by the advances in probabilistic video prediction and generative adversarial networks. The proposed model significantly outperforms prior work, scaling to high-resolution images (256x256) and long horizons (2500 frames).

---- Weaknesses ----

A weakness in the experimental setup is that the compared baselines are not controlled for the number of parameters. In particular, SVG-extend, which also works well on Kitti data, seems to contain the same number of parameters as the segmentation prediction network of the proposed model. It is also unclear how the baselines were tuned and what was the range of considered hypeparameters for all methods. More generally, it would be good for the paper to present some simple setting in which SVG-extend works well to analyze where hierarchical prediction is most important.

---- Additional comments ----

There is a number of minor inaccuracies in the paper:

The authors of Wichers’18 are cited as “Ruben Wichers, Nevan Villegas”. The real authors of this paper are Nevan Wichers and Ruben Villegas.

Page 2, “We employ the sequence model based on VAE (Denton & Fergus, 2018)”. Denton and Fergus did not invent VAEs, neither they invented sequential VAEs. You likely want to cite the original VAE papers (Kingma’14, Rezende’14), or the sequential VAE papers (Chung’15, Fraccaro’15). Somewhat strangely, none of these papers are cited elsewhere in the paper either. If you do want to cite Denton & Fergus, the sentence would need to be “We employ the sequence model based on VAE proposed by Denton & Fergus (2018)””.

Page 3, “we skip hidden representations of the encoder to the decoder at every time step during testing to handle longterm dynamics in structure”. Would be great to cite some papers that do also that, e.g. Villegas’17a or Finn’16.

“Extension to object boundary prediction”. This paragraph seems to be hastily written and full of incorrect statements. The G is not an autoencoder (and not a denoising autoencoder either). It predicts e from s, maybe just call it a boundary prediction network? The conditional GAN objective does not maximize p(e,s). In fact, p(s) is a constant. Instead, it tries to match p(e|s), but not by maximizing likelihood, but by minimizing an approximation to Jensen-Shannon divergence (see Goodfellow’14).

Eq (1) is not a variational lower bound for beta <> 1.

---

> ### Author Response · Authors · 2020-11-17
> **Rebuttal by Paper1712 Authors**
>
> We appreciate the insightful reviews.
>
> =====
>
> Q1. Start from the setting where SVG-extend works well to analyze where hierarchical prediction is most important.
> A1. We appreciate the comment. For reproducing SVG-extend, we employed the biggest one that fits into our GPUs since training the biggest and best-performing SVG-extend reported in [1] demands massive resources (32 TPUs). Then we opted for that model as our structure generator. In addition, we confirmed with the authors of SVG-extend [1] that our reproduced model nearly matches the performance of the biggest model.
>
> =====
>
> Q2. How the baselines were tuned and the range of considered hyper-parameters for all methods.
> A2. For all of the baselines, we reproduced the exact configurations and hyper-parameters reported in the papers or provided by the authors since (1) we believe the authors reported their best configurations and (2) our considered datasets (i.e, Human dancing, KITTI, and Cityscapes dataset) are the ones used in the original paper or barely deviate from the datasets considered by the baselines. In that regard, we consider it reasonable to expect the best performance with the original hyper-parameters.
>
> For instance, our Human Dancing dataset is composed of the single person dancing sequences under relatively static backgrounds, which resembles the Penn Action dataset. Therefore, we expect that the provided hyper-parameters of the hierarchical prediction model [2] on the Penn Action dataset might work in our dataset as well.
>
> =====
>
> Q3. The authors of Wichers’18 are cited as “Ruben Wichers, Nevan Villegas”. The real authors of this paper are Nevan Wichers and Ruben Villegas.
> A3. We appreciate the comment. We corrected the paper accordingly.
>
> =====
>
> Q4. Page 2, “We employ the sequence model based on VAE (Denton & Fergus, 2018)”.
> A4. We appreciate the comment. We clarified the sentence as suggested.
>
> =====
>
> Q5. Add citations for the paper utilizing skip-connections.
> A5. We appreciate the comment. We updated those citations.
>
> =====
>
> Q6. “Extension to object boundary prediction”. This paragraph seems to be hastily written and full of incorrect statements.
> A6. We appreciate the comment. We changed the term ‘denoising autoencoder’ into ‘conditional generator’ and also revised the description for conditional GAN objective.
>
> =====
>
> Q7. Eq (1) is not a variational lower bound for beta <> 1.
> A7. As augmenting the variational-lower bound with the 𝛽 (>=1) coefficient on KL divergence term is a common practice for modeling the data distribution [3], we adopted it into our approach. As you pointed out, Eq (1) is not a variational lower bound anymore if the 𝛽 is lower than 1, so we modified the term into ‘𝛽-VAE objective’. We will clarify this more in the revised paper.
>
> [1] Ruben Villegas, Arkanath Pathak, Harini Kannan, Dumitru Erhan, Quoc V Le, and Honglak Lee, High fidelity video prediction with large stochastic recurrent neural networks, In NeurIPS, 2019.
> [2] Ruben Villegas, Jimei Yang, Yuliang Zou, Sungryull Sohn, Xunyu Lin, and Honglak Lee, Learning to Generate Long-term Future via Hierarchical Prediction, In ICML, 2017b
> [3] Irina Higgins,  Loic Matthey,  Arka Pal,  Christopher Burgess,  Xavier Glorot,  Matthew Botvinick,Shakir  Mohamed,  and  Alexander  Lerchner,   beta-vae:  Learning  basic  visual  concepts  with  a constrained variational framework, In ICLR, 2017

---

> > ### Comment · AnonReviewer1 · 2020-11-20
> > **R1: the response is thorough and satisfactory**
> >
> > Thanks for the thorough response and revisions! The response resolves my initial questions. I also appreciate the new ablation. I believe the paper should be accepted.
> >
> > _Q1.  Number of parameters in the baselines_
> > The response convinces me that the comparison is adequate. However, it seems that the proposed method seems to have an advantage because it is able to train two models in two stages, therefore potentially using twice as many parameters. I understand that this might be outside of the scope of this work, but ideally a comparison should be conducted to an unsupervised two-stage method. For instance, a VQ-VAE can be trained in the first stage, and an SVG-extend can be trained on the VQ-VAE latents in the second stage, as in Rakhimov'20, Anonymous'20 (both concurrent work).
> >
> > Rakhimov'20, Latent Video Transformer
> > Anonymous'20, PREDICTING VIDEO WITH VQVAE
> >
> > _Q8._
> > Finally, it is important that the results of the proposed method be reproducible so that future work can meaningfully compare to them. If the training code cannot be released, it should be possible to release the inference code and the model weights.

---

### Official Review · AnonReviewer4 · 2020-10-26
**Some ablations missing**

**Rating:** 7
**Confidence:** 5

**Review:**

This paper proposes a VAE based hierarchical model for video prediction. The model employs recurrent model to predict intermediate representations (in the form of label maps) and these representations are mapped to pixel level information, i.e., videos. The paper presents an interesting idea of using representations that do not use any domain knowledge. The authors demonstrate the value of modeling temporal evolution of these representations which enables long term video prediction.

**Strengths**
+ The paper is clearly written and the implementation details are clearly described
+ The model outperforms relevant baselines quantitatively
+ The model is able to generate realistic and temporally consistent samples
+ The evaluation of the model is thorough and performs better consistently across challenging datasets.


**Weaknesses**
- How important is the warped optical flow in the video discriminator?
- How is $\tau'$ decided? Does it have any impact on the performance or computation requirements of the model?
- The role of $G_{edge}$ is unclear. An ablation study would be useful.
- Given the complexity of the task, it would be good to report the time required to train the models.

Overall, the paper presents impressive results on long-term video prediction and the evaluation of the paper is thorough. However, some ablations studies are required to convince the importance of each component used in the overall model. Therefore, my initial rating for this paper is 6.


=======================================**Post-rebuttal Comments**===============================


I appreciate the revisions and additional results reported by the authors. The authors have addressed the concerns raised by me in the revision. While I agree with R1 that novelty of the method is limited, after considering the reviews collectively, I believe this paper presents very impressive results given the fact that the problem is challenging. Therefore, I would like to improve my final rating to 7.

---

> ### Author Response · Authors · 2020-11-17
> **Rebuttal by Paper1712 Authors**
>
> We appreciate the insightful comments.
>
> =====
>
> Q1. Importance of the warped optical flow in the conditional video discriminator.
> A1. We appreciate the constructive comments. While the role of the optical flow based warping has been found to be essential for the visual quality of the generated frames [1], its importance in the conditional video discriminator has gotten less attention in the literature. However, as mentioned by the reviewer, we find that investigating the importance of the optical flow estimation in the conditional video discriminator could be beneficial since we can save computational and memory budgets consumed if its effect is found to be marginal. Due to the time limit, we will conduct this experiment during the upcoming round and try our best to revise the paper accordingly.
>
> =====
>
> Q2. Impact of τ’ on the performance and computation requirements.
> A2. The impact of varying τ’ was also explored by [1], where L is a corresponding term in the paper. According to the paper, it was reported that large τ’ increases computational costs with marginal performance improvement, while too small τ’ causes training instability. In our experiment, we simply chose τ’=5 (1) since we found that it was computationally affordable while not introducing training instability and (2) to keep consistency with the configuration of our structure generator that receives 5 frames as contexts.
>
> =====
>
> Q3. Role of G_edge.
> A3. We conducted the ablation study to understand the role of G_edge in our framework and updated the results in Section B.3.4. in the appendix. Quantitative results are summarized below where we compare the frame-wise evaluation (PSNR, SSIM, and VGG cosine similarity) of predicted sequences from the model with and without G_edge. Since the sequences of the Cityscapes dataset are relatively short (up to 30 frames), we observe marginal improvements in those metrics. However, we find that the generated frames contain more clear instance boundaries with G_edge, especially in crowded scenes. For qualitative results, please refer to Figure I in the appendix.
>
> |                	|&nbsp;  PSNR(↑)  &nbsp;| &nbsp; SSIM(↑) &nbsp; |&nbsp; CSIM(↑) &nbsp;|
> |---------------:	|:------:	|:-----:	|:-----:	|
> |    With G_edge 	| 20.427 	| 0.610 	| 0.938 	|
> | Without G_edge 	| 19.236 	| 0.602 	| 0.937 	|
>
> =====
>
> Q4. The time required to train the models.
> A4. Here we provide the computational and memory budgets for training our models and updated the table in the appendix accordingly.
>
> | Dataset                       	|&nbsp; &nbsp;&nbsp;    Model    | &nbsp; # GPU (V100 16GB) 	&nbsp; | Training Time 	|
> |-------------------------------	|:--------------:|:-----------------:	|:-------------:	|
> | KITTI & Human Dancing (64x64)   	|   SVG-extend   	|         1         	|    9 hours    	|
> | KITTI & Human Dancing (64x64)                              	|     Vid2Vid    	|         1         	|    12 hours   	|
> | KITTI & Human Dancing (256x256) 	|   SVG-extend   	|         4         	|    48 hours   	|
> | KITTI & Human Dancing (256x256)                              	|     Vid2Vid    	|         4         	|    48 hours   	|
> | Cityscapes (256x512)            	|   SVG-extend   	|         4         	|    48 hours   	|
> | Cityscapes (256x512)                              	| Edge Predictor 	|         1         	|    12 hours   	|
> | Cityscapes (256x512)                              	|     Vid2Vid    	|         8         	|    48 hours   	|
>
>
> [1] Ting-Chun Wang, Ming-Yu Liu, Jun-Yan Zhu, Guilin Liu, Andrew Tao, Jan Kautz, and Bryan Catanzaro, Video-to-video synthesis, In NeurIPS, 2018.

---

### Official Review · AnonReviewer2 · 2020-10-28

**Rating:** 6
**Confidence:** 4

**Review:**

###Summary###


The paper proposes the hierarchical video prediction model.
Specifically, the model first generates a sequence of semantic segmentation maps.
And then it generate a sequence of future frames corresponding to the semantic segmentation maps.
Since the sequence is generated stochastically, it can generate various futures given same condition.
With this hierarchical method, they could successfully generated thousands of future frames.


###Pros###


-
By using low-level information for modeling of dynamics, the proposed method generates realistic images not suffering from severe error amplification issue.



###Questions###

-
It is not clearly pointed why this method successfully generates long-term sequences.
Is it the result of learning the approximate posterior distribution of the following semantic segmentation map, or the result of the hierarchical generation mechanism?

-
Is the modeling motion only using the LSTM or conditional CNN inferior to this method? (deterministic one, no modeling for the approximate posterior distribution)

-
Regarding the equation 4, have you tried training the model without applying the teacher-forcing?
Which means, training with the generated semantic segmentation maps as the condition.

---

> ### Author Response · Authors · 2020-11-17
> **Rebuttal by Paper1712 Authors**
>
> We appreciate the insightful reviews.
>
> Q1. Key components for the successful long-term generation.
> A2. We appreciate the comment. As suggested by the reviewer, we conducted a more in-depth ablation study and isolated the key ingredients of the persistent prediction in our hierarchical framework. For detailed discussions, please find Section 5 in our revised paper. We provide a brief summarization below.
>
> (1) Stochastic estimation.
> To verify the impact of stochastic estimation, we implement the deterministic baseline by eliminating the stochasticity in the structure generator. We further compare our method with two deterministic baselines with (1) recurrent and (2) feedforward architectures. We found that deterministic models tend to seek the most likely local future and thus are prone to temporal misprediction and dramatic error propagations through time. In that regard, we find that the ability to anticipate diverse futures is one of the most important ingredients for persistent prediction since it can help the models recover from the bad minima and thereby make them robust to the errors in the prediction.
>
> (2) Recurrent estimation.
> Recurrent modules are beneficial for capturing temporal dependencies among structures and motions in different time-steps, which (a) helps improve prediction accuracy, (b) can alleviate the chance to fall into bad local minima, and (c) improve the overall generation quality.
>
> (3) Discretization of predicted label maps.
> We found that the performance of the structure generator decreases dramatically when it is modified to predict the soft labels (after softmax) of the segmentation map. Perhaps more surprisingly, we found that this modification makes the structure generator behave very similar to the one unrolling in RGB space. This result shows that predicting the *discrete* label map is the key to the persistent prediction.
>
> (4) Decomposed Hierarchical framework.
> By design, our structure generator is independent of the pixel-level errors (e.g. colors and textures) produced by the image generator, which along with the discretization process make our model robust to the mispredictions in pixels and structures. When we modify our method such that the predicted RGB frames are fed as input to the structure generator, we found a dramatic decrease in the performance.
>
> To summarize, we found that (1) using stochastic estimation, (2) unrolling the structure generator in a discrete space, and (3) decomposing the structure generation from image generation are the key ingredients for persistent prediction.
>
> If you have additional concerns, please let us know. We are happy to elaborate more.
>
> =====
>
> Q2. Modeling motion using SVG-extend versus deterministic LSTM/conditional CNN.
> A2. We performed an ablation study to investigate the effect of the stochastic estimation module in our structure generator, and revised the paper accordingly. Please find the details in Section 5 in the main paper.
>
> To summarize the results, we find that the stochastic estimation module is critical for persistent long-term prediction, where our ablated deterministic models (LSTM/CNN) get stuck in and cannot recover from bad local minima and thereby suffer from dramatic error propagations through time once the models have encountered them.
>
> =====
>
> Q3. Training conditional video generator without teacher forcing.
> A3. We appreciate the constructive comments. As mentioned by the reviewer, we can regularize the video generator (Vid2Vid) during training so that it learns to translate the generated label maps by the structure generator in a much realistic manner. Due to time and resource constraints, we will conduct this experiment in the upcoming round and try our best to update the results.

---

### Official Review · AnonReviewer3 · 2020-11-03
**Appealing visual results but with limited novelty and unspecific contribution.**

**Rating:** 5
**Confidence:** 5

**Review:**

Summary: This paper proposes a hierarchical framework for long-term video prediction. The structure is firstly predicted in the form of semantic map. It lies in a categorical structure space which is easier to predict. Then the authors translate the predicted semantic map to a real video sequence in a frame-by-frame manner. The proposed model is "surprisingly successful" for long-term video prediction, as claimed by the authors (thousands frames).

Pros:

+ clarity: This paper is overall easy to understand. The proposed method, which firstly predicts the categorical map and then translate  the map to video frame, is very straightforward and clear. The motivation behind such kind of design is convincing, i.e., to explicitly reduce the modelling difficulty of the prediction module.

+ significance: I have carefully checked the prediction results provided by the authors. I think the predicted video sequences are convincing and appealing, which are of >1000 frames. The video content does not freeze even after thousands frames. The visual quality on the cityscape dataset looks very promising.

Cons:

- originality: The proposed framework (first predicting semantic label then translating) is a simple extension compared to the previous work[1,2]. The model used in this work is a typical combination of LSTM and VAE, where the uncertainty is modeled with Gaussian Distribution and KL-divergence loss. For the second part, the auto-encoder based translation model, along with adversarial training scheme, is also commonly used in the image transfer task. Overall, the novelty of the proposed method is limited in my point of view.

- quality: Regarding this aspect, my major concern lies in the lack of the ablation study. In the experiment, the authors present sufficient and extensive results compared with other methods. However, the contribution of each module in the proposed method is completely unknown. Which part is the most important? Which part is the key to long-term video prediction? A comprehensive analysis about this part is highly recommended.

[`1] Predicting deeper into the future of semantic segmentation.
[2] Predicting future instance segmentation by forecasting convolutional features.

---

> ### Author Response · Authors · 2020-11-17
> **Rebuttal by Paper1712 Authors**
>
> We appreciate the constructive comments.
>
> Q1. Originality of the work
> A1. We appreciate the comments. We agree that our work is closely related to existing works. However, please note that long-term video prediction has remained an unsolved, open-problem over years, and this is the very first work demonstrating the success in this task: predicting *thousands* frames into the future in real-world videos. Please note that none of the existing works can produce a convincing future more than *dozens* of frames, despite extensive investigations in the past in various dimensions.
>
> Our contribution is providing a promising direction for ‘long-term’ prediction by carefully investigating missing components in the prior works: Compared to [1] proposing to maximize the capacity, we provide a counterargument that extrapolating in structure is more robust to error propagations. Compared to the feedforward method [5], we show that recurrent estimation is essential for capturing temporal variations in data. Compared to supervised [2, 3] and hierarchical methods [4], we show that stochastic estimation is critical for addressing deceptive bad local minima. Finally, compared to prior works exploiting the content of the last frame [1,5,6], we show that being able to handle the emerging concept is important.
>
> Putting it all together, we build a hierarchical approach that is simple and flexible. Since this is the first work addressing the video prediction at this scale, we also propose an evaluation protocol, shot-wise FVD, that can assess the prediction quality through time without ground-truth frames and intractable computation. We believe that our work will help future research to expand the scale of video prediction.
>
> =====
>
> Q2. Ablation study on the key components to the persistent long-term prediction of the proposed model.
> A2. We appreciate the comment. As suggested by the reviewer, we conducted a more in-depth ablation study and isolated the key ingredients of the persistent prediction in our hierarchical framework. For detailed discussions, please find Section 5 in our revised paper. We provide a brief summarization below.
>
> (1) Stochastic estimation
> To verify the impact of stochastic estimation, we implement the deterministic baseline by eliminating the stochasticity in the structure generator. We further compare our method with two deterministic baselines with (1) recurrent and (2) feedforward architectures. We found that deterministic models tend to seek the most likely local future and thus are prone to temporal misprediction and dramatic error propagations through time. In that regard, we find that the ability to anticipate diverse futures is one of the most important ingredients for persistent prediction since it can help the models recover from the bad minima and thereby make them robust to the errors in the prediction.
>
> (2) Recurrent estimation
> Recurrent modules are beneficial for capturing temporal dependencies among structures and motions in different time-steps, which (a) helps improve prediction accuracy, (b) can alleviate the chance to fall into bad local minima, and (c) improve the overall generation quality.
>
> (3) Discretization of predicted label maps
> We found that the performance of the structure generator decreases dramatically when it is modified to predict the soft labels (after softmax) of the segmentation map. Perhaps more surprisingly, we found that this modification makes the structure generator behave very similar to the one unrolling in RGB space. This result shows that predicting the *discrete* label map is the key to the persistent prediction.
>
> (4) Decomposed Hierarchical framework
> By design, our structure generator is independent of the errors produced by the image generator (e.g. colors and textures), which along with the discretization process make our model robust to the mispredictions in pixels and structures. When we modify our method such that the predicted RGB frames are fed as input to the structure generator, we found a dramatic decrease in the performance.
>
> To summarize, we found that (1) using stochastic estimation, (2) unrolling the structure generator in a discrete space, and (3) decomposing the structure generation from image generation are the key ingredients for persistent prediction.
>
> If you have additional concerns, please let us know. We are happy to elaborate more.
>
> [1] Villegas et al., High fidelity video prediction with large stochastic recurrent neural networks
> [2] Luc et al., Predicting future instance segmentation by forecasting convolutional features
> [3] Luc et al., Predicting deeper into the future of semantic segmentation
> [4] Villegas et al., Learning to Generate Long-term Future via Hierarchical Prediction
> [5] Bhattacharyya et al., Bayesian prediction of future street scenes using synthetic likelihoods
> [6] Denton et al., Stochastic video generation with a learned prior

---

### Decision · Program_Chairs · 2021-01-07
**Final Decision**

**Decision:**

Accept (Poster)

**Comment:**

This paper proposes a new implementation of a previously proposed two-stage process for video prediction: first predict future segmentation maps, then map them to video frames. Combined with other advances in video prediction and image generation, this simple idea is shown empirically to work very well, producing video predictions up to many hundreds of frames into the future in real stochastic settings with unprecedented quality. Strong ablation studies over the course of the review process further serve to confirm the value of various design choices involved in the implementation.